# Beyond Binary Rewards: Training LMs to Reason about Their Uncertainty

**Mehul Damani**[*]   **Isha Puri**[*]   **Stewart Slocum**   **Idan Shenfeld**   **Leshem Choshen**
**Yoon Kim**   **Jacob Andreas**
Massachusetts Institute of Technology

## Abstract

When language models (LMs) are trained via reinforcement learning (RL) to generate natural language reasoning chains, their performance improves on a variety of difficult question answering tasks. Today, almost all successful applications of RL for reasoning use binary reward functions that evaluate the correctness of LM outputs. Because such reward functions do not penalize guessing or low-confidence outputs, they often have the unintended side-effect of degrading calibration and increasing the rate at which LMs generate incorrect responses (i.e. "hallucinate″) in other problem domains. This paper describes **RLCR** (Reinforcement Learning with Calibration Rewards), an approach to training reasoning models that jointly improves accuracy and calibrated confidence estimation. During RLCR, LMs generate both predictions and numerical confidence estimates after reasoning. They are trained to optimize a reward function that augments a binary correctness score with a Brier score—a scoring rule for confidence estimates that incentivizes calibrated prediction. We first prove that this reward function (or any analogous reward function that uses a bounded, proper scoring rule) yields models whose predictions are both accurate and well-calibrated. We next show that across diverse datasets, RLCR substantially improves calibration while maintaining strong accuracy on both in-domain and out-of-domain evaluations—outperforming both ordinary RL training and classifiers trained to assign post-hoc confidence scores. While ordinary RL hurts calibration, RLCR improves it. Finally, we demonstrate that verbalized confidence can be leveraged at test time to improve accuracy and calibration via confidence-weighted scaling methods. Our results show that explicitly optimizing for calibration can produce more generally reliable reasoning models.

## 1 Introduction

Many recent advances in research on language models (LMs) have been driven by *reasoning models*—LMs trained via reinforcement learning (RL) to 'think out loud' in natural language before answering questions, achieving state-of-the-art performance on challenging tasks like math and programming  (Guo et al., 2025).

The standard approach to reasoning training (often referred to as **reinforcement learning with verifiable rewards**, or **RLVR**) performs RL with a simple binary correctness reward: $R_{\text{correctness}}(y, y^*) = \mathbb{1}_{y \equiv y^*}$, where $\equiv$ checks whether the model's output $y$ matches ground-truth answer $y^*$. While simple and effective for improving accuracy, this reward comes with a critical limitation: it rewards models equally whether they are confidently correct or merely guessing, and penalizes identically whether they abstain or produce incorrect answers. This incentivizes overconfident guessing.

Consistent with this concern, studies have shown that even when initially well-calibrated, models tend to become overconfident following RL training (Choshen et al., 2019; Leng et al., 2025). Reasoning models, in particular, tend to exhibit worsened calibration and increased hallucination rates compared to base models, especially when trained with reward signals that emphasize only correctness (Kirichenko et al., 2025; Yao et al., 2025; OpenAI, 2025). This is a critical limitation in high-stakes domains such as healthcare or law, where models must not only be accurate but also communicate uncertainty when appropriate (Omar et al., 2024).

---

[*]Equal Contribution. Correspondence to mehul42@mit.edu.

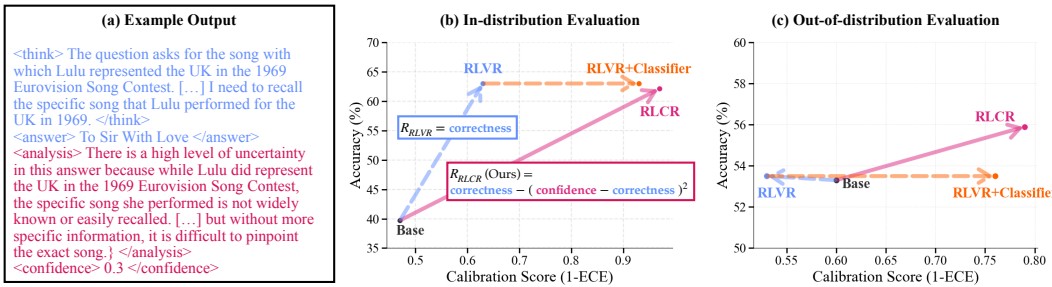

Figure 1: **(a):** Sample chain-of-thought from a model trained with RLCR, using `<think>`, `<answer>`, `<analysis>`, and `<confidence>` tags. **(b)** On in-domain evaluation tasks, RLCR improves on standard reasoning training (RLVR) and even slightly outperforms a combination of RLVR and a dedicated classifier trained to predict RLVR correctness. **(c)** When evaluating generalization to novel tasks, RLCR improves both accuracy and calibration, while other methods leave accuracy unchanged and sometimes harm calibration. All results shown are for HotpotQA, see Section 4 for more results.

This paper aims to address these limitations by answering two questions:

(1) Can reasoning models be optimized for both correctness and calibration?

(2) Can the contents of reasoning chains themselves improve calibration?

We approach these questions through the lens of statistical decision theory, specifically the theory of **proper scoring rules**. Given a predictor that produces an output $y$ and a confidence $q$, a proper scoring rule is minimized when $q$ reflects the true probability that $y$ will agree with a ground-truth outcome $y^*$ (Gneiting & Raftery, 2007). A canonical example is the **Brier score** (Brier, 1950): $R_{\text{Brier}}(y, q, y^*) = -(q - \mathbb{1}_{y \equiv y^*})^2$. Proper scoring rules are widely used in forecasting (Waghmare & Ziegel, 2025), but have seen little application in training LLMs with RL.

Our approach, **RLCR (reinforcement learning with calibration rewards)**, involves a modified version of reasoning training that encourages models to reason about both task correctness and uncertainty. To do so, we simply train models to output both answers $y$ and (verbalized) confidence scores $q$, optimizing a combined reward function:

$$\begin{aligned} R_{\text{RLCR}}(y, q, y^*) &= R_{\text{correctness}}(y, y^*) + R_{\text{Brier}}(y, q, y^*) \\ &= \mathbb{1}_{y \equiv y^*} - (q - \mathbb{1}_{y \equiv y^*})^2 . \end{aligned} \tag{1}$$

We show that this approach has several appealing theoretical and empirical properties:

- RLCR provably incentivizes both accuracy and calibration: $R_{\text{RLCR}}$ is maximized when models output the answer most likely to be correct, along with a calibrated estimate of their probability of success. In other words, $R_{\text{RLCR}}$ is maximized by LM outputs $(y, q)$ for which $y = \arg\max_{y'} p(y' \equiv y^*)$, and $q = p(y \equiv y^*)$, where $p$ denotes the true underlying probability distribution over correctness labels. More generally, we show that an analogous objective can be constructed whenever a *bounded*, proper scoring rule is used for the calibration term. Notably, log-likelihood, though a proper scoring rule, is unbounded and can incentivize models to output incorrect answers (Section 3).

- In experiments on factual question answering and mathematical reasoning tasks, RLCR matches the task accuracy of RLVR while substantially improving calibration, on in-domain problems, reducing expected calibration error from $0.37 \rightarrow 0.03$ on HotpotQA (Yang et al., 2018) and $0.26 \rightarrow 0.10$ on a collection of math datasets (Section 4).

- RLCR improves calibration on out-of-domain tasks: where RLVR substantially worsens calibration generalization to new domains, RLCR improves calibration, outperforming the RLVR model, base model and a predictor equipped with a second model fine-tuned only to output confidence scores.

- Verbalized confidence can be incorporated into test-time scaling, improving ensembling and best-of-$N$ methods: This may be attributed to the fact that RLVR also improves the *coherence* of model predictions across samples: when multiple reasoning chains and predictions are generated for a given question, RLCR reduces the variance in confidence scores across reasoning chains that

lead to the same answer, and reduces the frequency with which models assign high confidence to contradictory answers (Section 4.4).

Together, these results show that existing reasoning training methods can be straightforwardly modified to additionally optimize for calibration, and that this in turn improves their accuracy, robustness, and scalability.

## 2 PRELIMINARIES

Let $\pi_\theta$ be a language model that maps from prompts $x \in X$ to outputs $y \in Y$, perhaps preceded by a natural language reasoning chain, with $x$, $y$ and reasoning chains all represented as strings. Given a dataset of prompt–output pairs $D = \{(x_i, y_i^*)\}$ (e.g. questions and ground-truth answers) and a reward function $R : Y \times Y \to \mathbb{R}$ that compares predicted to ground-truth outputs, our goal is to improve LM outputs by optimizing:

$$\arg\max_\theta \; \mathbb{E}_{(x,y^*)\sim D, \; y\sim\pi_\theta(\cdot|x)} \, R(y, y^*) \,. \tag{2}$$

**Reinforcement learning with verifiable rewards (RLVR)**   When training reasoning models, a standard choice of $R$ is the binary correctness reward:

$$R_{\text{correctness}}(y, y^*) = \mathbb{1}_{y\equiv y^*} \,, \tag{3}$$

where $\mathbb{1}_{y\equiv y^*} \in \{0, 1\}$ is the indicator function that evaluates whether $y$ is correct, i.e. equivalent (perhaps modulo formatting details) to $y^*$.

**Proper scoring rules**   Often, we want predictors that output not only an answer $y$, but some scalar measure $q$ of confidence in this answer.[1] A **scoring rule** measures the quality of a confidence estimate. In the case of modeling binary outcomes (e.g. our confidence that a given answer $y$ is correct), a scoring rule is a function $S : \mathbb{R} \times \{0, 1\} \to \mathbb{R}$ that maps a confidence estimate $q$ and an outcome $a$ to a scalar score. A scoring rule is called **proper** if its expected value is minimized by confidence scores that match the true outcome probability:

$$\mathbb{E}_{a\sim p(a)} S(p(a), a) \leq \mathbb{E}_{a\sim p(a)} S(q, a) \;\; \forall q \,. \tag{4}$$

Here, $p$ denotes the true underlying probability distribution over correctness labels. Perhaps the most familiar example of a proper scoring rule is the log-loss:

**Logarithmic score:**   $\qquad S(q, a) = -a \log q - (1 - a) \log(1 - q) \,. \tag{5}$

But many other examples exist, including

**Brier score:**   $\qquad S(q, a) = (a - q)^2 \,, \tag{6}$

**Spherical score:**   $\qquad S(q, a) = -\dfrac{qa + (1 - q)(1 - a)}{\sqrt{q^2 + (1 - q)^2}} \,. \tag{7}$

What all these scores have in common is the property that they are minimized when confidences $q$ match the true probability $p(a = 1)$.

## 3 METHOD

The main idea behind our approach is to train language models via reinforcement learning with a reward that incentivizes *both* correctness and calibration, by combining a standard correctness reward with a reward based on the Brier score. In this approach, models are first prompted to produce reasoning chains that produce both answers and confidences estimates (as in Fig. 1a). They are then trained to optimize:

$$R_{\text{RLCR}}(y, q, y^*) = \mathbb{1}_{y\equiv y^*} - (q - \mathbb{1}_{y\equiv y^*})^2 \,. \tag{8}$$

Intuitively, this reward incentivizes correctness but penalizes models when they output incorrect answers with high confidence or correct answers with low confidence.

---

[1]It is sometimes even more useful to train models that can place a complete *distribution* over a large set of possible answers $y$. But for very large answer spaces or expensive predictors—like language models performing chain-of-thought reasoning—enumerating and scoring all possible answers is generally impractical. This paper mainly focuses on models that generate one answer and one confidence score, though see Section 4.6 for one way of using this approach to generate and score multiple answers.

It is not immediately obvious that this reward function incentivizes desired LM behavior—because it involves a tradeoff between accuracy (the first term) and calibration (the second), we might worry that models will learn to output answers certain to be wrong in order to obtain a small calibration loss. But in fact, the calibration term in Eq. (8) comes at no cost in accuracy:

**Theorem 1.** *Suppose, for any prediction $y$, that the success indicator $\mathbb{1}_{y \equiv y^*}$ is distributed as Bernoulli($p_y$). Then $R_{RLCR}$ in Eq. (8) satisfies two properties:*

1. **Calibration incentive.** *For any $y$, the expected reward $\mathbb{E}_{\mathbb{1}_{y \equiv y^*}} R_{RLCR}(y, q, y^*)$ is maximized when $q = p_y$.*

2. **Correctness incentive.** *Among all calibrated predictions $(y, p_y)$, expected reward is maximized by the prediction whose success probability $p_y$ is greatest.[2]*

Proof is given in Appendix A.

An important property of Theorem 1 is that we cannot replace the Brier term $(q - \mathbb{1}_{y \equiv y^*})^2$ with *any* proper scoring rule—for example the log loss $\mathbb{1}_{y \equiv y^*} \log q + (1 - \mathbb{1}_{y \equiv y^*}) \log(1 - q)$ does not incentivize correctness. However an analogous version of Theorem 1 exists for any *bounded* proper scoring rule satisfying $S(p, 1) - S(p, 0) < \lambda$ for some $\lambda$.

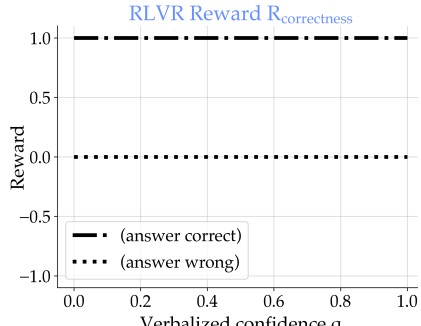

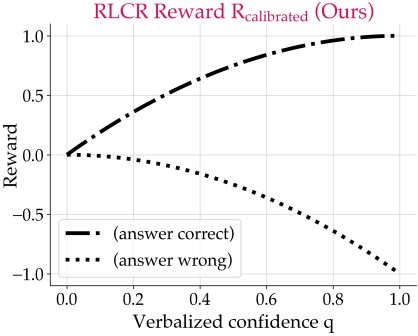

Figure 2: **(a):** RLVR focuses solely on correctness, which can incentivize guessing. **(b):** RLCR uses a calibrated reward that jointly optimizes for correctness and calibration.

## 4 EXPERIMENTS

Our main experiments aim to evaluate how RLCR empirically changes the accuracy and calibration of LMs, both in "in-domain" evaluations on the task used for RL, and "out-of-domain" evaluations on other question-answering tasks. Additional experiments evaluate interactions between RLCR and other test-time reasoning paradigms, and the extent to which RLCR causes LM predictions to become more coherent *across* predictions.

### 4.1 EXPERIMENTAL SETUP

**Training Details** We use Group Relative Policy Optimization (GRPO) (Shao et al., 2024) as the base RL algorithm with some modifications (see Section B.2). We use the Qwen2.5-7B base model, part of the Qwen family popularly used in RL tasks (Hu et al., 2025; Gandhi et al., 2025). Following recent work on RL training for LM reasoning (Hu et al., 2025; Guo et al., 2025), we initialize RL from the base model and do not use any KL regularization.

**Methods** We evaluate the following methods:

1. **Base:** The base pre-trained model. We use *Qwen2.5-7B Base* in our experiments. We prompt the model to output both answers and confidences, detailed in Section B.4.

2. **RLVR:** Initialized from the base model and trained using $R_{correctness}$ with `<think>` and `<answer>` tags. During evaluation, the model is also prompted to output a verbalized confidence.

---

[2]Note that statement of the problem does not distinguish between epistemic and aleatoric uncertainty about success. Obviously, once $y$ has been predicted, the outcome of evaluation is fully determined, and the objective probability that $y \equiv y^*$ is either 0 or 1. But an information- or computation-constrained predictor may still possess subjective uncertainty.

3. **RLVR + BCE Classifier:** A confidence classifier trained on outputs from the RLVR model. Specifically, given problems, solution CoTs (from RLVR), and correctness labels $(x, y, \mathbb{1}_{y \equiv y^*})$, we train a confidence classifier $f_\theta(x, y)$ using the binary cross-entropy (BCE) loss:

$$\mathcal{L}_{\text{BCE}}(\theta) = -\mathbb{E}_{(x,y,\mathbb{1}_{y \equiv y^*})} \left[ \mathbb{1}_{y \equiv y^*} \log f_\theta(x, y) + (1 - \mathbb{1}_{y \equiv y^*}) \log(1 - f_\theta(x, y)) \right] \quad (9)$$

The classifier is initialized from *Qwen2.5-7B Base* and is thus highly expressive. This approach is expensive, as it requires training and inference with two large models.

4. **RLVR + Brier Classifier:** Instead of using binary cross-entropy (BCE) loss to train a classifier, we use mean squared error (MSE), which allows more direct optimization of the Brier score:

$$\mathcal{L}_{\text{Brier}}(\theta) = \mathbb{E}_{(x,y,\mathbb{1}_{y \equiv y^*})} \left[ (f_\theta(x, y) - \mathbb{1}_{y = y^*})^2 \right] \quad (10)$$

5. **RLVR + Probe:** Given final-layer embedding $\phi(x, y)$ of the RLVR model, we train a linear probe to predict confidence. In Eq. (9), we replace the fine-tuned LM with a linear model: $f_\theta(x, y) = \log \sigma(\theta^\top \phi(x, y))$, where $\sigma(\cdot)$ denotes the sigmoid function.

6. **Answer Probability:** We generate outputs using RLVR, extract tokens enclosed within <answer> tags, and compute their average probability: $\text{AnswerProb}(y) = \frac{1}{|\mathcal{A}|} \sum_{t \in \mathcal{A}} P_\theta(y_t \mid y_{<t}, x)$

   Here, the set $\mathcal{A} \subseteq \{1, \ldots, T\}$ denotes the token positions that appear between the <answer> tags. $P_\theta(y_t \mid y_{<t}, x)$ represents the model's probability of generating token $y_t$.

7. **RLCR (ours):** Initialized from base model and trained using $R_{\text{RLCR}}$.

**Evaluation Metrics** We use the following evaluation metrics:

1. **Accuracy (↑):** A measure of performance.

2. **Area under ROC curve (AUROC) (↑):** Measures how well confidence scores distinguish correct from incorrect answers, treating correctness as a binary label and averaging TPR/FPR (true and false positive rates) over all thresholds. $\text{AUROC} = \int_0^1 \text{TPR}(\text{FPR}^{-1}(t)) \, dt$

3. **Brier Score (↓):** Squared difference between confidence and ground truth. Brier Score $= \frac{1}{N} \sum_{i=1}^{N} (q_i - \mathbb{1}_{y_i \equiv y_i^*})^2$

4. **Expected Calibration Error (ECE) (↓):** Calibration metric that groups confidences into bins and computes difference between the average correctness and confidence. $\text{ECE} = \sum_{m=1}^{M} \frac{|B_m|}{N} |\text{acc}(B_m) - \text{conf}(B_m)|$, where $M$ is the number of bins, $B_m$ is the set of samples in bin $m$, and $N$ is the number of samples. We use $M = 10$.

**Evaluation datasets** We evaluate RLCR on benchmarks highlighting distinct sources of uncertainty, including ambiguous evidence, obscure facts, and complex reasoning. HotPotQA (Yang et al., 2018) tests calibration under incomplete or distracting evidence, while SimpleQA (Wei et al., 2024) and TriviaQA (Joshi et al., 2017) probe overconfidence on obscure factual knowledge; GPQA (Rein et al., 2024), Math500 (Hendrycks et al., 2021), GSM8K (Cobbe et al., 2021), and Big-Math (Albalak et al., 2025) assess calibration in complex, multi-step or scientific reasoning, where uncertainty accumulates across steps. CommonsenseQA (Talmor et al., 2019) examines confidence in ambiguous, implicit reasoning scenarios.

## 4.2 HOTPOTQA

**Dataset** We use a modified HotPotQA distractor dataset (Yang et al., 2018) with multi-hop questions and 10 paragraphs (2 relevant, 8 distractors). To test uncertainty reasoning, *HotPotQA-Modified* removes 0, 1, or both relevant paragraphs, creating varying information completeness. See Appendix M for an example. The dataset is evenly split across these conditions, with 8 paragraphs per example. We train on 20,000 examples and use exact string match to compute correctness.

**Results** On the in-distribution HotpotQA distractor dataset (Table 1), RL-trained models outperform off-the-shelf baselines in multi-hop accuracy, with RLCR being comparable to RLVR in accuracy, showing that the calibration term does not hurt performance. While both the base and RLVR remain overconfident and poorly calibrated, our method and the classifiers achieve substantially better calibration, with RLCR slightly ahead. The *Answer Probability* baseline performs poorly, as the

**(a) Models Trained on HotpotQA**

| Method | HotpotQA | | | | O.O.D. Averaged | | | |
|---|---|---|---|---|---|---|---|---|
| | Acc. (↑) | AUROC (↑) | Brier (↓) | ECE (↓) | Acc. (↑) | AUROC (↑) | Brier (↓) | ECE (↓) |
| Base | 39.7% | 0.54 | 0.53 | 0.53 | 53.3% | 0.54 | 0.41 | 0.40 |
| RLVR | | 0.50 | 0.37 | 0.37 | | 0.50 | 0.46 | 0.46 |
| RLVR + BCE Classifier | | 0.66 | 0.22 | 0.07 | | 0.58 | 0.27 | 0.24 |
| RLVR + Brier Classifier | **63.0%** | 0.65 | 0.22 | 0.09 | 53.9% | 0.60 | 0.32 | 0.33 |
| RLVR + Probe | | 0.55 | 0.24 | 0.10 | | 0.53 | 0.38 | 0.38 |
| Answer Probability | | **0.72** | 0.36 | 0.36 | | 0.60 | 0.42 | 0.42 |
| RLCR (**ours**) | 62.1% | 0.69 | **0.21** | **0.03** | **56.2%** | **0.68** | **0.21** | **0.21** |

**(b) Models Trained on Big-Math**

| Method | Math | | | | O.O.D. Averaged | | | |
|---|---|---|---|---|---|---|---|---|
| | Acc. (↑) | AUROC (↑) | Brier (↓) | ECE (↓) | Acc. (↑) | AUROC (↑) | Brier (↓) | ECE (↓) |
| Base | 56.1% | 0.56 | 0.40 | 0.39 | 47.8% | 0.53 | 0.46 | 0.45 |
| RLVR | | 0.47 | 0.28 | 0.26 | | 0.52 | 0.49 | 0.49 |
| RLVR + BCE Classifier | | **0.78** | 0.15 | 0.10 | | 0.55 | 0.34 | 0.33 |
| RLVR + Brier Classifier | **72.9%** | 0.78 | 0.15 | 0.10 | **52.5%** | 0.57 | 0.28 | 0.27 |
| RLVR + Probe | | 0.65 | 0.19 | 0.13 | | 0.53 | 0.33 | 0.30 |
| Answer Probability | | 0.52 | 0.26 | 0.26 | | 0.52 | 0.44 | 0.43 |
| RLCR (**ours**) | 72.7% | 0.67 | 0.17 | 0.10 | 50.9% | 0.60 | 0.28 | 0.25 |
| SFT+RLCR (**ours**) | 72.2% | **0.78** | **0.14** | **0.08** | 43.8% | **0.66** | **0.24** | **0.18** |

Table 1: **Accuracy and calibration metrics for models trained on HotpotQA and Big-Math. Best values bolded. Dataset-specific results in Appendix K.**
(a) **Performance on HotpotQA and** 6 **out-of-distribution (O.O.D.) datasets.** RLCR achieves competitive accuracy and significantly outperforms all baselines in calibration, especially on O.O.D. datasets, demonstrating the benefits of jointly optimizing accuracy and calibration.
(b) **Performance on Math and** 5 **out-of-distribution (O.O.D.) datasets.** Math results are averaged over 3 datasets: Math-500, GSM8K and Big-Math. SFT+RLCR variant achieves the best calibration across both in-distribution and O.O.D. settings. However, this comes at the cost of reduced generalization accuracy, possibly due to catastrophic forgetting. RLCR offers a stronger trade-off in O.O.D. settings, maintaining competitive accuracy while improving calibration.

model typically commits to an answer during CoT reasoning, inflating output confidence. Fig. 11 shows that both correctness and calibration reward for RLCR increase smoothly during training.

Next, we evaluate experiments across six out-of-distribution datasets (TriviaQA, SimpleQA, MATH500, GSM8K, CommonsenseQA, GPQA). We find that RL training on HotpotQA does not improve accuracy OOD, with the base model performing comparably to RL-trained models. However, RLVR worsens calibration relative to the base model, whereas RLCR achieves substantial gains over all baselines across calibration metrics while maintaining (or slightly improving) on task accuracy. We hypothesize that better calibration generalization of RLCR could be due to:

1. **Chain-of-thought reasoning about uncertainty** Reasoning about uncertainty can improve calibration by allowing reflection on confidence, in line with recent work (Yoon et al., 2025).

2. **Training dynamics of RL** During RL training, the model's confidence analysis and scores have to constantly adapt to the model's improving task performance. This non-stationarity might lead to more robust learning and better generalization.

3. **Shared representations** Using a single model for both solution generation and calibration allows the calibration task to leverage internal representations used by the solution generating process.

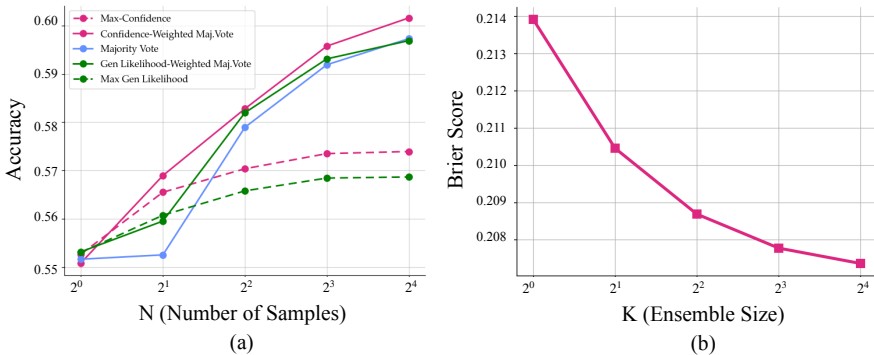

Figure 3: **Test-time scaling curves. (a) Accuracy vs Number of Samples (N)**. Accuracy improves for all methods with increasing compute. *Confidence-weighted majority vote* outperforms both vanilla *majority vote* and *max-confidence*, highlighting complementary benefits of combining voting with confidence scores. **(b) Brier Scores vs Ensemble Size (K).** Here we evalute the effect of applying test-time scaling to *confidence estimation alone*, resampling multiple analyses (blue text in Fig. 1). Calibration improves (although modestly) as the size of the analysis ensemble grows.

### 4.3 MATH

**Training Details**  We use a subset of the Big-Math (Albalak et al., 2025) dataset, containing 15,000 problems selected using criteria defined in Section B.1. We compute correctness using *math-verify*.[3] To enhance the quality of uncertainty reasoning, we also train a variant with a lightweight SFT warmup phase. We generate solutions from the base model on 500 examples and use Deepseek-R1 to produce uncertainty analyses for them. Further details in Section B.2.

**Results**  On Math benchmarks (averaged over GSM8K, Math, and Big-Math), all RL methods improve accuracy significantly over the base model. SFT+RLCR achieves the best calibration, slightly surpassing the classifiers, while base and RLVR remain poorly calibrated. Out-of-distribution (TriviaQA, SimpleQA, CommonsenseQA, GPQA, HotPotQA), the accuracies of RLCR and RLVR are marginally better than the base model, but surprisingly the accuracy of the SFT+RLCR model drops significantly, possibly due to catastrophic forgetting induced by SFT warmup. Despite this, SFT+RLCR achieves the strongest calibration. Overall, RLCR offers a stronger trade-off in O.O.D. settings, maintaining accuracy while matching or outperforming all baselines on calibration.

### 4.4 CAN VERBALIZED CONFIDENCES BE USED FOR TEST-TIME SCALING?

We next evaluate whether confidence scores from RLCR can be incorporated into test-time scaling algorithms to yield improvements in both accuracy and calibration.

**Accuracy**  Test-time scaling methods such as best-of-N or majority vote are widely used to boost model performance by aggregating multiple responses. Given $N$ responses $y_1, y_2, ...y_N$ and a reward model $r(x, y)$, **best-of-N** selects the response with highest reward: $y_{\text{chosen}} = \arg\max_{\{y_1,...,y_N\}\sim p(\cdot|x)} R(x, y_i)$. Similarly, **majority vote** selects the most frequent answer among $N$ responses. Prior work has also explored **weighted majority vote**, where the voting is weighted using reward model scores.

**Our key insight is that the verbalized confidence $q$ output by the model can serve as an effective proxy reward** (Taubenfeld et al., 2025). If a model is well-calibrated, then selecting responses with higher confidence will naturally lead to an increase in performance. This insight leads to two simple algorithms: select the response with the highest verbalized confidence from the set of $N$ candidates (**max-confidence**), or weight each vote by its confidence score (**confidence-weighted majority vote**). These algorithms are the confidence-scored analogues to Best-of-N and weighted majority vote. Importantly, both approaches do not require any additional supervision or external reward models.

---

[3]https://github.com/huggingface/Math-Verify

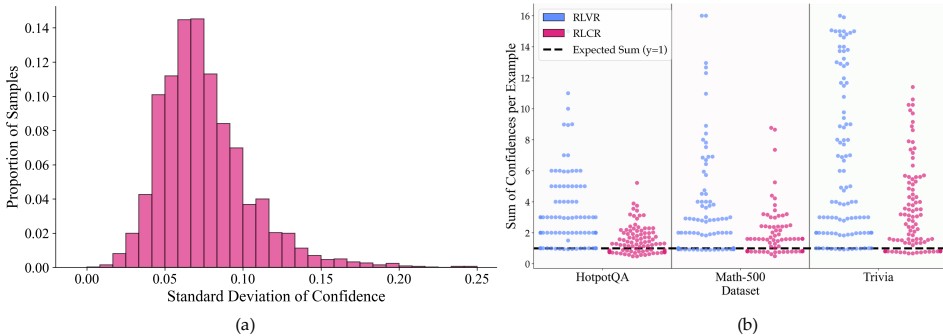

Figure 4: **(a): Distribution of standard deviation in confidence across multiple uncertainty reasoning chains for the same solution/answer.** Most samples exhibit low deviation, indicating that the model's confidence estimates are self-consistent. (b) **Swarm plot of confidence sums across 3 datasets**. RLCR consistently remains closer to the ideal sum of 1. Nonetheless, overconfidence remains, suggesting room for further improvement.

To contextualize these methods against likelihood-based selection, which is always computable for any open model, we additionally compare against two likelihood-based baselines - **max generation likelihood**, which performs Best-of-N using average sentence likelihood $R = p(y|x)/|y|$ as the reward score, and **generation likelihood-weighted majority vote**, which weights votes by these likelihood scores. We evaluate these approaches using the RLCR model (trained on Hotpot) for generation, plotting average accuracy across the 7 datasets used in Table 1. As shown in Fig. 3a, accuracy consistently improves with more samples, with confidence weighted majority vote outperforming vanilla majority voting, max-confidence, and both likelihood baselines. These gains highlight how calibration can directly underpin test time scaling - better calibrated confidence estimates lead to more accurate aggregated predictions.

**Calibration**  In our structured RLCR CoT, models first output a solution, followed by an analysis and confidence score. To improve confidence scores for a fixed answer $y$, we sample $K$ analysis CoTs $z_1, \ldots, z_K \sim p(\cdot \mid x, y)$, each producing a verbalized confidence $q_i$. We then ensemble these confidences to obtain the aggregated confidence estimate $\bar{q} = \frac{1}{K} \sum_i q_i$. Figure 3b plots Brier score (averaged over 7 datasets) as a function of ensemble size $K$. We observe that calibration improves as the ensemble size grows, though the absolute gains are relatively modest. This reflects the fact that for most questions, there is low *"uncertainty about uncertainty"*, so averaging does not substantially alter the estimate. Nonetheless, ensembling provides a lightweight mechanism for reducing residual noise, especially on harder questions where confidence variability is greater.

### 4.5 ARE VERBALIZED CONFIDENCES SELF-CONSISTENT?

**Intra-solution coherence**  A desirable property of uncertainty-aware reasoning is that a model should assign consistent confidence estimates when generating multiple uncertainty reasoning chains for the same answer. That is, the model should have low *"uncertainty about uncertainty"*. Given a fixed answer $y$, we sample $K$ analysis CoTs $z_1, \ldots, z_K \sim p(\cdot \mid x, y)$, where each chain produces a verbalized confidence score $q_i$. Ideally, these confidence scores should exhibit low variability.

Figure 4a plots the standard deviation across seven datasets, using analysis CoTs generated by the RLCR model trained on HotpotQA. Most samples have low standard deviation, suggesting that the model's confidence estimates are generally consistent.

**Inter-solution consistency**  For tasks where answers are *mutually exclusive*—i.e., only one answer is correct per instance—it is desirable that the model distributes its confidence across distinct answers such that the total confidence is less than or equal to 1, with equality holding when the set of answers is exhaustive. Let a model generate $N$ responses $\{y_i\}_{i=1}^N$ with associated confidence scores $\{q_i\}_{i=1}^N$, where $q_i \in [0, 1]$. Let $\mathcal{A} = \{a_1, \ldots, a_K\}$ denote the set of $K \leq N$ unique answers among the $y_i$. The mean confidence assigned to answer $a_k$ is:

$$\bar{q}_k = \frac{\sum_i \mathbb{1}_{y_i \equiv a_k} \cdot q_i}{\sum_i \mathbb{1}_{y_i \equiv a_k}} \tag{11}$$

| Models Trained on HotpotQA | | | | | | | | |
|---|---|---|---|---|---|---|---|---|
| **Method** | **HotpotQA** | | | | **O.O.D. Averaged** | | | |
| | **Acc.** | **Tokens** | **Brier** | **ECE** | **Acc.** | **Tokens** | **Brier** | **ECE** |
| | ($\uparrow$) | ($\downarrow$) | ($\downarrow$) | ($\downarrow$) | ($\uparrow$) | ($\downarrow$) | ($\downarrow$) | ($\downarrow$) |
| RLCR (**ours**) | 62.1% | 249 | **0.21** | **0.03** | 56.2% | 300 | **0.21** | **0.21** |
| RLCR w/o Analysis | 61.7% | 113 | 0.23 | 0.09 | **56.5**% | 179 | 0.26 | 0.26 |
| RLVR w/ Analysis | 62.3% | 224 | 0.35 | 0.34 | 52.6% | 221 | 0.41 | 0.39 |
| RLVR | **63.0**% | **92** | 0.37 | 0.37 | 53.9% | **142** | 0.46 | 0.46 |

Table 2: **Calibration ablation evaluating the contributions of (i) calibration-aware reward learning and (ii) explicit uncertainty reasoning**. Adding uncertainty analysis improves calibration for both RLCR and RLVR, while removing analysis from RLCR still preserves significant calibration benefits. Together, these results indicate that both components independently contribute to improved calibration, and the strongest performance is achieved when both are used together.

For a model to give consistent answers, we desire: $\sum_{k=1}^{K} \bar{q}_k \leq 1$.

Fig. 4b shows a swarm plot of predicted confidence sums across three representative datasets. On the in-distribution HotpotQA dataset, RLCR's confidence sums cluster tightly around 1, indicating well-calibrated belief distribution. For out-of-distribution datasets, both RLCR and RLVR exhibit overconfidence, with sums exceeding 1, though RLCR remains significantly closer to the ideal. This reflects RLCR 's improved calibration, yet highlights room for improvement, particularly OOD.

### 4.6 WHAT DRIVES CALIBRATION GAINS?

RLCR differs from RLVR along two key axes: (1) RLCR's reward function directly incentivizes calibrated confidence estimates, and (2) RLCR models are prompted to explicitly reason about uncertainty. To isolate the contribution of these components, we ablate the uncertainty-reasoning portion of the CoT prompt and evaluate the following four variants on the HotpotQA-trained models:

1. **RLCR:** Vanilla RLCR, as presented in Table 1.
2. **RLCR w/o Analysis:** The RLCR model evaluated *without* uncertainty reasoning. At inference time, the model is instructed to output only `<think>`, `<answer>`, and `<confidence>`. The model is instructed to not perform any uncertainty reasoning.
3. **RLVR:** Vanilla RLVR, as presented in Table 1.
4. **RLVR w/ Analysis:** The RLVR model evaluated *with* uncertainty reasoning, using the identical analysis prompt used for training/evaluating RLCR (see long RLCR prompt in Appendix C).

The results in Table 2 reveal several key findings.

**(1) Explicit uncertainty reasoning improves calibration for *both* RLCR and RLVR.** Across both training and OOD settings, the variants with uncertainty reasoning achieve lower Brier and ECE than their no-analysis counterparts. This is consistent with prior work (Yoon et al., 2025) and is further corroborated by the classifier ablations in Appendix G.

**(2) Reward-based calibration is substantially more effective than prompting alone.** Although RLVR w/ Analysis improves over vanilla RLVR, it remains far behind both RLCR variants, including RLCR w/o Analysis. Thus, prompting a model to reason about uncertainty provides only modest gains compared to explicitly training with a calibrated reward.

**(3) RLCR w/o Analysis nearly matches RLVR in accuracy and token cost, while dramatically improving calibration.** On HotpotQA, RLCR w/o Analysis uses a similar number of tokens as RLVR (113 vs. 92) and achieves comparable accuracy (61.7% vs. 63.0%), yet its calibration is far superior (ECE: 0.09 vs. 0.37).

**Overall.** Both components—explicit analysis and calibration-aware reward learning—contribute to improved calibration. When efficiency is paramount, RLCR w/o Analysis offers a drop-in alternative that preserves accuracy and token efficiency while significantly improving calibration over RLVR.

## 5 RELATED WORK

Building reliable reasoning models requires not only high accuracy but also calibrated uncertainty—a property that standard RL objectives often erode (Achiam et al., 2023), leaving models over-confident (Xiong et al., 2024) and prone to hallucination (Jaech et al., 2024). We survey four strands of prior work on confidence estimation in LLMs:

**Post-hoc verbalizations** prompt models to state their confidence after answering (Xiong et al., 2024; Yang et al., 2024; Tanneru et al., 2024; Lin et al., 2022). Lin et al. (2022) fine-tune GPT-3 to predict confidence given a question and answer, using empirical accuracy as the target label. Xiong et al. (2024) find that LMs exhibit overconfidence when verbalizing their confidence. Tian et al. (2023) finds that RLHF models' verbalized confidence scores are better calibrated than their conditional probabilities. Mei et al. (2025) find that even reasoning models also suffer from overconfidence and can become even more overconfident with deeper reasoning. Similarly, Kirichenko et al. (2025) introduce AbstentionBench and show that LMs struggle to abstain appropriately, with reasoning fine-tuning often degrading abstention performance. Positively, Yoon et al. (2025) and Mei et al. (2025) find that reasoning models can improve calibration by introspection and slow thinking.

**Sampling-based methods** use response agreement (e.g., majority vote or best-of-$N$) as a proxy for confidence, but are costly and require clear ground truth (Kang et al., 2025). Aichberger et al. (2025) estimate uncertainty by generating diverse, plausible responses and measuring their consistency. Kuhn et al. (2023) propose semantic entropy, a sampling-based method that leverages linguistic invariances to better estimate uncertainty in natural language generation.

**Internal probing** extracts confidence from model features like token probabilities (Gupta et al., 2024), offering fine-grained scores but lacking generality. Kadavath et al. (2022) prompt language models to output "true" or "false" and use their probability as a proxy for the model's confidence. Mielke et al. (2022) train a LM to to generate responses conditioned on confidence estimates provided by an external probe. Fadeeva et al. (2024) propose a token-level uncertainty method that leverages internal model signals to fact-check claims and detect hallucinations. Azaria & Mitchell (2023) train a classifier on hidden layer activations to detect the truthfulness of statements based on the LLM's internal state. Orgad et al. (2025) show that internal representations encode rich, token-level truthfulness signals, which can detect and categorize errors beyond what is reflected in the output.

**RL-based methods** train models to output calibrated verbal confidences with RL. Stengel-Eskin et al. (2024) introduce a speaker-listener framework, where the speaker's rewards are determined by the listener's inferred confidence in the speaker's responses. Leng et al. (2025) introduce verbalized confidence scores in reward model training. Most closely related to our work, Xu et al. (2024) and Stangel et al. (2025) train LMs with RL using proper scoring rules as reward functions. Xu et al. (2024) adopt the Brier score, while Stangel et al. (2025) use a clipped version of the log loss. While effective, these approaches optimize solely for calibration, which can inadvertently harm task accuracy—particularly in larger models that may reward hack by outputting deliberately incorrect answers with zero confidence to achieve perfect calibration rewards. Furthermore, both methods are developed and evaluated exclusively on non-reasoning tasks. In contrast, our method incentivizes both correctness and calibration, and is evaluated on both reasoning and non-reasoning benchmarks.

## 6 CONCLUSION

We show that incorporating proper scoring rules into RL, via an objective we call RLCR, enables reasoning models to improve both accuracy and calibration. Our approach trains models to reason about and verbalize uncertainty, preserving task performance while significantly improving calibration in- and out-of-distribution. We demonstrate that reasoning about uncertainty improves calibration, and that our method improves the self-consistency of confidence, and improves with test-time scaling. However, there remains significant room for improvement—even after RLCR, out-of-domain calibration error is often high in an absolute sense, and models may still assign high confidence to multiple contradictory answers. Nevertheless, these results suggest a path toward reasoning systems that are not only accurate, but reliably reason about and communicate uncertainty.

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

# A  PROOF OF THEOREM 1

We consider a slightly more general family of reward functions than in the main paper body. As above, we assume that predictors produce a response $y$, a confidence $q \in [0, 1]$; we now assume that scores depend additionally on an arbitrary binary correctness signal $c$, distributed according to some $p_y := p(c = 1 \mid y)$.

We consider all reward functions of the form:

$$R(c, q) = \lambda c - S(q, c)$$

where $S(q, c)$ is a scoring rule and $\lambda > 0$ is a scalar reward for producing the correct answer.

We define the expected reward of choosing a confidence $q$ for a given response $y$ as:

$$V(y, q) = \mathbb{E}_c\, R(c, q) = \lambda p_y - [p_y S(q, 1) + (1 - p_y) S(q, 0)] \ .$$

**Lemma 1** (Calibration incentive). *For any response $y$, the expected reward $V(y, q)$ is maximized at $q = p_y$ if and only if $S(q, c)$ is a proper scoring rule.*

*Proof.* The correctness term $\lambda p_y$ in the reward function does not depend on $q$, so

$$\arg\max_q \ V(y, q) = \arg\max_q \ -p_y S(q, 1) + (1 - p_y) S(q, 0) = \arg\min_q \ \mathbb{E}_c\, S(q, c) \ .$$

But by a scoring rule is proper by definition if $\mathbb{E}_{c \sim \text{Bernoulli}(p_y)}\, S(q, c)$ is minimized by $q = p_y$, so the statement of the lemma follows immediately. $\qquad\square$

**Lemma 2** (Correctness incentive). *Consider $y, y'$ with associated success probabilities $p_y \geq p_{y'}$. Then $V(y, p_y) \geq V(y', p_{y'})$ if and only if*

$$S(p, 1) - S(p, 0) \leq \lambda \quad \text{for all } p \in [0, 1] \ .$$

*Proof.* First, define:
$$W(p) = \lambda p - p S(p, 1) + (1 - p) S(p, 0) \ ,$$

Note that $V(y, p_y) = W(p_y)$, and $W(p)$ represents the maximum reward attainable for any $y$ with associated success probability $y_p$. Thus, to verify the statement of the lemma, it suffices to show that $W(p) \geq W(p')$ if and only if $p \geq p'$, which in turn is equivalent to showing that $W(p)$ is nondecreasing in $p$.

We first compute the derivative of $W$:

$$W'(p) = \lambda - S(p, 1) + S(p, 0) - p S'(p, 1) - (1 - p) S'(p, 0) \ .$$

From the Savage-Dawid representation (Gneiting & Raftery, 2007) of proper scoring rules, there exists some non-negative weight function $\omega(t) \geq 0$ such that:

$$S'(p, 1) = -(1 - p) \cdot \omega(p) \ , \qquad S'(p, 0) = p \cdot \omega(p) \ .$$

Substituting this in:

$$\begin{aligned} W'(p) &= \lambda - S(p, 1) + S(p, 0) + p \cdot (1 - p) \cdot \omega(p) - (1 - p) \cdot p \cdot \omega(p) \\ &= S(p, 1) - S(p, 0) \ . \end{aligned}$$

Thus the derivative of $W$ is non-negative ($W$ is nondecreasing) if and only if $S(p, 1) - S(p, 0) \leq \lambda$ for $p \in [0, 1]$. $\qquad\square$

Then the main theorem statement in Section 3 follows immediately from these two lemmas:

*Proof of Theorem 1.* First observe that $R_{\text{RLCR}}$ satisfies the conditions of Lemma 2 with $\lambda = 1$ and $S(q, \mathbb{1}_{y \equiv y^*}) = (q - \mathbb{1}_{y \equiv y^*})^2$, we have $\max_p\, S(p, 1) - S(p, 0) = 1$. Then condition 1 (calibration) follows from Lemma 1 and the fact that the Brier score is a proper scoring rule, and condition 2 (correctness) follows from Lemma 2 and the boundedness of $S(p, 1) - S(p, 0)$. $\qquad\square$

**Corollary 1.** *Let $S(q, c)$ be a strictly proper scoring rule.*

1. *If $S(p,1) - S(p,0)$ is* bounded*, then there exists a finite $\lambda > 0$ such that the reward function $R(c,q) = \lambda c - S(q,c)$ satisfies the correctness condition:*

$$S(p,1) - S(p,0) \leq \lambda \quad \text{for all } p \in [0,1]$$

   *and thus jointly incentivizes calibration and correctness.*

2. *If $S(p,1) - S(p,0)$ is* unbounded*, then for any finite $\lambda > 0$, there may exist some $y \geq y'$ such that $W(p_y) < W(p_{y'})$, and $R_{RLCR}$ prefers $(y', p_{y'})$ to $(y, p_y)$.*

**Examples.** The Brier score is bounded: $S(p,1) = (1-p)^2$, $S(p,0) = p^2$, so:

$$S(p,1) - S(p,0) = 1 - 2p \leq 1 \quad \text{for all } p \in [0,1]$$

Thus, the condition holds for $\lambda = 1$.

In contrast, the logarithmic score is unbounded:

$$S(p,1) = -\log p, \quad S(p,0) = -\log(1-p), \quad S(p,1) - S(p,0) = \log\left(\frac{1-p}{p}\right) \to \infty \quad \text{as } p \to 0$$

So no finite $\lambda$ can satisfy the condition.

## B  EXPERIMENTAL SETUP

### B.1  TRAINING DATASETS

**HotpotQA-Modified:** We use a modified version of the HotPotQA distractor dataset, which contains factual questions requiring multi-hop reasoning. (Yang et al., 2018). Each example in this setting presents ten paragraphs, only two of which contain the information necessary to answer the question; the remaining eight paragraphs include closely related but irrelevant details. Consequently, solving this task requires the model to identify and reason over the pertinent passages. To more strongly develop uncertainty reasoning capability, we construct a new dataset, *HotPotQA-Modified*, in which we systematically remove either 0, 1, or both of the key paragraphs required to answer each question. This modification introduces varying levels of informational completeness that the model must reason over. We distribute questions across three equal groups: one-third have no relevant paragraphs (0/8), one-third have 1 relevant paragraph (1/7), and one-third have both relevant paragraphs (2/6). Each question consistently contains 8 total paragraphs. Our training dataset consists of 20,000 examples. We measure correctness using exact string match.

**Big-Math Digits:** We use Big-Math (Albalak et al., 2025), a large, curated training dataset for RL containing over 250,000 math problems, including questions from benchmarks such as Math and GSM8K. To ensure an appropriate range of difficulty, we retain problems for which the LLaMA-8B solve rate (provided in the dataset) is between 0-70%. We also found that verifier noise can be significant in Math datasets and can cause training instability. To reduce verifier noise, we further restrict the dataset to problems with numerical answers, enabling near-perfect automatic verification. Our final training set consists of 15,000 problems. We compute correctness using *math-verify*.

### B.2  ADDITIONAL TRAINING DETAILS

Following Turtel et al. (2025), we remove the standard deviation division in the advantage, which might help with learning on examples where there are extreme miscalibrations. We use the BNPO loss function, which aggregates token level losses using the number of active tokens in the local training batch (Xiao et al., 2025). We generate 32 responses per prompt with a temperature of 0.7, and use an effective batch size of 2048. Experiments were conducted on both NVIDIA A100 and H100 GPUs (and we observed consistent results across hardware types). We use a constant learning rate with warmup of 1e-06 for HotpotQA and 5e-06 for Math. We use a warmup ratio 0.1. For Hotpot, we set a maximum completion length of 1536 while for Math, we use a completion length of 4096. Hotpot requires significantly less reasoning, and using a smaller completion length helped improve training time. We do 1 epoch of training. For training *RLCR*, we use the *Long RLCR* system prompt for Hotpot and the *Simple RLCR* prompt for Math (the long version did not provide additional benefit on Math). We use the *Simple Generation* prompt for *RLVR*. All prompts in Appendix C.

**Format Reward:** We use a format reward to encourage adherence to the structured format shown in Fig. 1. In RLVR, models must format their output in <think> and <answer> tags. In RLCR, in addition to <think> and <answer> tags, we require an <analysis> tag to enclose uncertainty reasoning and a <confidence> tag for verbalized confidence. A valid response must contain all these tags in the correct order. Both format and calibration rewards are weighted equally.

**SFT Warmup in Math:** While RL directly on the base model improves calibrated reward, the uncertainty analyses produced in Math remain qualitatively generic, often lacking reasoning tied to specific solution steps (See Appendix M for an example). To improve their quality, we train a variant with a lightweight SFT warmup phase before RL to obtain higher quality uncertainty analyses. We generate solutions from the base model on 500 examples and prompt Deepseek-R1 with the *Expert SFT Prompt* to produce uncertainty analyses for them. We then perform SFT with the <think> and <answer> obtained from the base model, appended with the <analysis> obtained from Deepseek-R1. Note that we do not ask Deepseek-R1 to output confidence scores.

### B.3  EVALUATION DATASETS

We run evaluation on a large number of datasets:

1. **HotPotQA (Distractor):** We use 1000 validation examples from the original HotpotQA distractor dataset. We slightly modify the dataset and remove 2 non-relevant paragraphs from each question. Thus, each question has 8 paragraphs with both supporting paragraphs present. We measure correctness using exact-match (Yang et al., 2018).

2. **HotPotQA-Modified:** We evaluate on 500 held-out validation examples. We measure correctness using exact-match.

3. **TriviaQA:** We use 2000 examples from the validation set of the TriviaQA dataset (Joshi et al., 2017). We use the no-context split to purely test factual accuracy.We evaluate using LLM-as-a-judge.

4. **SimpleQA:** We use the full SimpleQA dataset consisting of 4326 factual questions (Wei et al., 2024). We evaluate using LLM-as-a-judge.

5. **Math-500** We use the popular MATH-500 dataset, which contains a subset of problems from the original MATH dataset (Hendrycks et al., 2021). We evaluate using *math-verify*, a mathematical expression evaluation system released by huggingface.

6. **GSM8K:** We use the test set (1319 problems) of the popular Grade School Math 8K dataset (Cobbe et al., 2021). We evaluate using *math-verify*.

7. **Big-Math-Digits:** We evaluate on 1000 held-out validation examples. We evaluate using *math-verify*.

8. **CommonSenseQA:** We use the validation set (1220 problems) of the CommonsenseQA dataset (Talmor et al., 2019), a multiple-choice question answering dataset that requires different types of commonsense knowledge to predict the correct answers. We evaluate using LLM-as-a-judge.

9. **GPQA:** We use the GPQA main dataset containing 448 multiple-choice questions written by experts in biology, physics, and chemistry (Rein et al., 2024). We evaluate using LLM-as-a-judge.

## B.4 Evaluation Details

All models are evaluated with temperature 0. For all datasets except Math and GSM8K, we use a maximum token budget of 4096. The system prompt for evaluation and the pipeline to extract answer and confidence scores varies slightly based on the method we are evaluating:

1. **RLCR (ours):** RLCR models use `<think>`, `<answer>`, `<analysis>` and `<confidence>` tags. They are evaluated with the same system prompts they are trained on. We extract their answer from `<answer>` tag and their confidence from `<confidence>` tag.

2. **RLVR:** RLVR models use the `<think>` and `<answer>`. It is evaluated with the same system prompt and we extract their answer from the `<answer>` tag. To obtain their verbalized confidence, we append *"Thinking time ended. My verbalized confidence in my answer as a number between 0 and 100 is equal to"* to their generated output.

3. **Classifier/Probe:** Both methods are conditioned on the question and the *RLVR* model's generation (solution and answer). These methods thus use *RLVR* model as a generator and their reported accuracies in the result tables are equal.

4. **Base:** The base model is not good at instruction following and is prompted with a simpler system prompt (*Simple Confidence Prompt*) that guides it to use `<think>`, `<answer>` and `<confidence>` tags. In case no valid confidence can be extracted, we append *"Thinking time ended. My verbalized confidence in my answer as a number between 0 and 100 is equal to"* to their output and call them again to extract confidence.

For all methods, if we are unable to extract a valid answer from the `<answer>` tags, we append *"Thinking time ended. My final answer is"* to their output and call them again. The main goal of these custom pipelines is to be able to fairly extract an answer and a confidence level and minimize cases where incorrect formatting adversely affects performance. Note that because they are trained with format rewards, both the RL-trained models are nearly perfect in adhering to the desired format and require minimal interventions. However, the base model benefits from this full extraction pipeline.

**Importantly, once answers and confidences are extraced, all methods are evaluated identically and based on the dataset, exact-match, LLM-as-a-judge or math-verify is used.**

**LLM-as-a-judge:** We use Llama-3.1-8B-Instruct with temperature set to 0 as our judge. The judge is provided with the question, the ground truth answer and the answer extracted from the evaluation pipeline. It is prompted to respond with "YES" or "NO" based on the correctness of the answer. As the datasets we evaluate have short and objective answers, we do not condition the judge on the thinking traces which can add biases.

## C  SYSTEM PROMPTS

---

**Long RLCR Prompt**

"A conversation between User and Assistant. The user asks a question, and the Assistant solves it. The assistant first thinks about the reasoning process in the mind, provides the user with the final answer, then analyzes its confidence about the solution and then provides the user with its confidence level. The confidence level is a number between 0 and 1 (inclusive) enclosed within `<confidence> </confidence>` tags. The final answer is enclosed between `<answer> </answer>` tags. The analysis about confidence and uncertainty is enclosed within `<analysis> </analysis>` tags. The assistant should reason about its confidence in the solution and its uncertainty in the solution within these tags. Here are some guidelines for the analysis: 1. Your task is to point out things where the model could be wrong in its thinking, or things where there might be ambiguity in the solution steps, or in the reasoning process itself.
2. You should not suggest ways of fixing the response, your job is only to reason about uncertainties.
3. For some questions, the response might be correct. In these cases, It is also okay to have only a small number of uncertainties and then explicitly say that I am unable to spot more uncertainties.
4. Uncertainties might be different from errors. For example, uncertainties may arise from ambiguities in the question, or from the application of a particular lemma/proof.
5. If there are alternate potential approaches that may lead to different answers, you should mention them.
6. List out plausible uncertainties, do not make generic statements, be as specific about uncertainties as possible.
7. Enclose this uncertainty analysis within `<analysis> </analysis>` tags.
The final format that must be followed is :  `<think>` reasoning process here `</think> <answer>` final answer here `</analysis> <analysis>` analysis about confidence and uncertainty here `</analysis> <confidence>` confidence level here (number between 0 and 1) `</confidence>` )

---

**Simple RLCR Prompt**

A conversation between User and Assistant. The user asks a question, and the Assistant solves it. The Assistant first thinks about the reasoning process in the mind, provides the user with the final answer, then analyzes its confidence about the solution and provides the user with its confidence level. The confidence level is a number between 0 and 1 (inclusive) enclosed within `<confidence> </confidence>` tags. The final answer is enclosed between `<answer> </answer>` tags. The analysis about confidence and uncertainty is enclosed within `<analysis> </analysis>` tags. The Assistant should reason about its confidence in the solution and its uncertainty in the solution within these tags. The final format that must be followed is: `<think>` reasoning process here `</think><answer>` final answer here `</answer><analysis>` analysis about confidence and uncertainty here `</analysis><confidence>` confidence level here (number between 0 and 1) `</confidence>`

---

.

> **Simple Confidence Prompt**
>
> A conversation between User and Assistant. The user asks a question, and the Assistant solves it. The assistant first thinks about the reasoning process in the mind and analyzes its confidence about the solution and then provides the user with the final answer as well as its confidence level. The confidence level is a number between 0 and 1 (inclusive) enclosed within `<confidence> </confidence>` tags. The final answer is enclosed between `<answer> </answer>` tags. The final format that must be followed is : `<think>` reasoning process here `<//think><answer>` final answer here `</answer>` `<confidence>` confidence level here (number between 0 and 1) `</confidence>`.

.

> **RLVR Prompt**
>
> A conversation between User and Assistant. The user asks a question, and the Assistant solves it. The assistant first thinks about the reasoning process in the mind and then provides the user with the answer. The reasoning process and answer are enclosed within `<think> </think>` and `<answer> </answer>` tags, respectively, i.e., `<think>` reasoning process here `</think><answer>` answer here `</answer>`.

.

> **Expert SFT Prompt**
>
> You are given a question and a solution to it. You have to verify if the solution is correct and enclose your verification reasoning within `<analysis> </analysis>` tags. Your analysis should be a minimum of 300 characters and should sequentially go through the thinking solution step by step. Here are the guidelines for your analysis:
> 1. Your analysis should also be in 'I' form as if you wrote the solution and are now verifying it.
> 2. Your goal is not to solve the problem but instead to verify if the steps in the presented solution are correct.
> 3. If there are ambiguities in the solution steps or if a step introduces uncertainty, you should mention it in the analysis.
> 4. Go through the solution sequentially in a step-by-step manner.
> 5. The analysis should be 300 characters minimum.
> 6. Enclose this uncertainty analysis within `<analysis> </analysis>` tags.

## D   LLM USAGE

The authors made limited use of ChatGPT to refine wording for clarity. It was not used for research ideation, related work retrieval, or substantive content generation.

# E  COMPARING LOG SCORE AND BRIER SCORE

## E.1  INTUITION

Logarithmic score and Brier scores are both widely used proper scoring rules. Lemma 1 showed that combining proper scoring rules with correctness rewards preserves the calibration incentive. Models are incentivized to output true correctness probability $q = p_y$ for both log and Brier scores. Using this, we can write out the expected reward when the model outputs answer $y$ and honestly reports confidence $q = p_y$:

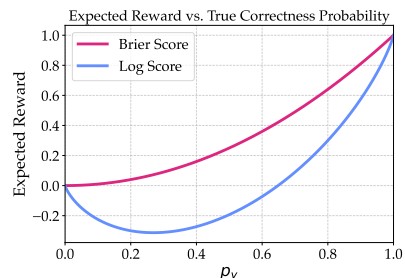

Expected Reward vs. True Correctness Probability

$$\mathbb{E}_{\text{brier}}[R(y)] = p_y\big(1 - (1 - p_y)^2\big) + (1 - p_y)(-p_y^2) \tag{12}$$

$$\mathbb{E}_{\log}[R(y)] = p_y(1 + \log p_y) + (1 - p_y)\log(1 - p_y). \tag{13}$$

Figure 5: Expected Reward vs correctness probability $p_y$.

Fig. 5 plots the expected reward as a function of $p_y$. While the Brier-based expected reward is strictly increasing in $p_y$, the expected reward under the log score is not. This means that, when using log score combined with correctness rewards, there exist ranges of $p_y$ where the model receives higher expected reward by reporting answers with *lower* true correctness probability. Intuitively, the log score's calibration term can outweigh the correctness bonus, creating a non-monotonic reward landscape that can discourage the model from preferring more accurate answers.

## E.2  TOY EXPERIMENT

**Environment Setup.** To empirically validate this phenomenon, we construct a simple $K$-arm prediction task. For each instance, Nature samples a probability vector

$$\mathbf{p} = (p_1, \ldots, p_K), \qquad p_i \geq 0, \sum_{i=1}^{K} p_i = 1,$$

which is *unknown* to the model. A sequence of $N$ IID draws

$$x_1, \ldots, x_N \sim \mathbf{p}, \qquad x_t \in \{1, \ldots, K\},$$

is revealed to the model in the prompt. The value of $N$ is drawn uniformly from

$$N \sim \text{Unif}\{0, 1, 2, 3, 4, 5\}.$$

The model's task is to predict the next draw

$$x_{N+1} \sim \mathbf{p},$$

by outputting an arm index $\hat{y} \in \{1, \ldots, K\}$ along with a confidence value $q \in [0, 1]$. The model is free to output an invalid answer $\hat{y} = -1$ with $q = 0$ if it wishes to abstain. We use $K = 5$ throughout. A sample task instance is shown below.

.

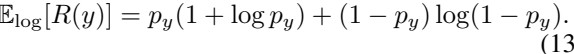

> **Toy Arm Task Instance**
>
> There are 5 arms, each arm has a different probability of being sampled. You are given some draws from this 5-arm distribution. Your task is to predict which arm the next draw will be from. You are also free to output -1 with confidence 0 if you are really unsure.
> Observed draws (in order): 1, 1, 0.
> Number of observed draws shown: 3
> Answer with the arm index (0-4) that you predict for the next draw.

Since the observation sequence is short and $\mathbf{p}$ varies arbitrarily across tasks, this setting induces substantial aleatoric uncertainty. The Bayes-optimal prediction conditioned on the observed data is the arm with highest empirical frequency,

$$\hat{y}^{\star} = \arg \max_{i \in \{1,...,K\}} \sum_{t=1}^{N} \mathbf{1}\{x_t = i\}.$$

This environment is intentionally designed so that the model is often highly uncertain about the correct arm. Referring to Fig. 5, we observe that under the logarithmic scoring rule, the expected reward in such low-confidence regimes is maximized by outputting an intentionally incorrect answer with confidence 0, rather than attempting a good-faith prediction.

**Training Setup.** We train RLCR-Brier and RLCR-Log on $10,000$ examples from the above dataset. We use the Qwen-2.5-7B model and use the same training configuration as Section B.2. For simplicity, we do not ask the model to do any uncertainty reasoning.

**Results.**

Fig. 6 presents evaluation results on the Toy Arm task. Sample outputs from the two models are also shown below. The empirical findings closely match our theoretical predictions. Under the logarithmic scoring rule, the model rapidly converges to a degenerate policy: it outputs the invalid arm $(-1)$ with reported confidence $q = 0$ for nearly all tasks. This strategy yields the *maximum* possible expected reward under the log score in the low–$p_y$ regime, and consequently the model achieves an almost perfect calibration metric. However, because the model never predicts a valid arm, its accuracy on the underlying task collapses to 0.

| Method | Acc. | Brier | ECE |
|---|---|---|---|
| RLCR-Brier (**ours**) | **34.4%** | 0.22 | 0.02 |
| RLCR-Log | 0 | **0.00** | **0.00** |

Figure 6: Performance on the Toy Arm Task dataset. RLCR-Log collapses to degenerate solution, while RLCR-Brier achieves non-trivial accuracy and calibration.

In contrast, RLCR-Brier behaves qualitatively differently. Because the expected Brier reward is strictly increasing in the true correctness probability, collapsing to the invalid low-confidence answer is strongly suboptimal. Models trained with RLCR-Brier therefore continue to make genuine predictions about the next arm, achieving both non-trivial accuracy and stable calibration.

.

> **RLCR-Brier Sample Output**
>
> `<think>` Analyzing the frequency of each arm `</think><answer>` 1 `</answer>`
> `<confidence>` 0.25 `</confidence>`

.

> **RLCR-Log Sample Output**
>
> `<think>` Based on the given information, there is no data to predict the next draw. `</think><answer>` -1 `</answer>` `<confidence>` 0`</confidence>`

### E.3 EXPERIMENTS ON HOTPOT

We also evaluate the two scoring rules in a realistic QA setting. To do so, we train an RLCR-Log model using the same training configuration as the RLCR models used in our main HotpotQA results (see Table 1). Table 3 reports a direct comparison between RLCR-Brier and RLCR-Log; the RLCR-Brier and RLVR numbers are copied over from the main results table.

Overall, RLCR-Log performs slightly worse than RLCR-Brier on both accuracy and calibration. However, unlike the toy bandit setting, we do not observe any evidence of reward hacking: the RLCR-Log model maintains reasonable accuracy and does not collapse to degenerate predictions. This suggests that even though using log score can lead to reward hacking, for many datasets hacking

| Method | HotpotQA | | | | O.O.D Averaged | | | |
|---|---|---|---|---|---|---|---|---|
| | Acc. | AUROC | Brier | ECE | Acc. | AUROC | Brier | ECE |
| RLVR | **63.0**% | 0.50 | 0.37 | 0.37 | 53.9% | 0.50 | 0.46 | 0.46 |
| RLCR-Brier (**ours**) | 62.1% | **0.69** | **0.21** | **0.03** | **56.2**% | **0.68** | **0.21** | **0.21** |
| RLCR-Log | 59.5% | 0.68 | 0.22 | 0.07 | 53.6% | 0.67 | 0.22 | **0.21** |

Table 3: **Comparison of RLCR-Brier and RLCR-Log on HotpotQA and** 6 **out-of-distribution (O.O.D.) datasets**. RLCR-Brier marginally outperforms RLCR-Log in both accuracy and calibration.

might not happen in practice. We believe the emergence of hacking is dependent on both the data distribution as well as the model size.

# F GENERALIZATION TO OTHER MODELS

## F.1 OLMO-2

| Method | HotpotQA | | | | O.O.D Averaged | | | |
|---|---|---|---|---|---|---|---|---|
| | Acc. | AUROC | Brier | ECE | Acc. | AUROC | Brier | ECE |
| Base | 16.4% | 0.55 | 0.78 | 0.79 | 47.8% | 0.56 | 0.48 | 0.48 |
| RLVR | **61.7%** | 0.51 | 0.38 | 0.38 | **50.8**% | 0.53 | 0.48 | 0.48 |
| RLCR (**ours**) | 61.3% | **0.58** | **0.24** | **0.09** | 49.3% | **0.65** | **0.22** | **0.20** |

Table 4: **Results of RLCR and RLVR training using OlMo-2-7B-Instruct**. RLCR has comparable accuracy to RLVR, while significantly outperforming it in calibration.

**Setup:** To assess how well the RLCR framework generalizes across model families, we train RLCR and RLVR variants starting from the OlMo-2-7B-Instruct model (OLMo et al., 2024) on our HotpotQA-Modified dataset.

**Results:** Table 4 reports the final performance. The overall trends are consistent with our main findings: RLCR and RLVR achieve similar accuracy, and both substantially improve accuracy over the base model. RLCR also delivers markedly better calibration than both RLVR and the base model.

## F.2 QWEN3

| Method | HotpotQA | | | | O.O.D Averaged | | | |
|---|---|---|---|---|---|---|---|---|
| | Acc. | AUROC | Brier | ECE | Acc. | AUROC | Brier | ECE |
| Base | 61.1% | 0.53 | 0.35 | 0.34 | 62.8% | 0.59 | 0.28 | 0.28 |
| RLVR | **62.7%** | 0.53 | 0.36 | 0.36 | 65.5% | 0.62 | 0.28 | 0.29 |
| RLCR (**ours**) | 61.8% | **0.58** | **0.28** | **0.23** | **65.6%** | **0.71** | **0.17** | **0.17** |

Table 5: **Results of RLCR and RLVR training using Qwen-3-8B**. RLCR maintains accuracy while substantially improving calibration in both in-distribution and OOD settings.

We further assess performance on the newly released Qwen3 model family. Under in-distribution evaluation, RLCR matches RLVR in accuracy and surpasses it on AUROC, Brier, and ECE. When averaged across our OOD benchmarks, RLCR continues to outperform RLVR on AUROC, Brier, and ECE, with accuracy remaining effectively equivalent.

# G    DOES REASONING IMPROVE CALIBRATION?

Recent work has shown that CoT reasoning can be unfaithful, with generated CoTs that do not influence their final answers (Chen et al., 2025). This raises the possibility that uncertainty analysis may not meaningfully inform the verbalized confidence score. To test this, we train two classifiers on *HotPotQA-Modified*:

1. **Baseline classifier:** Trained on RLVR outputs (these contain no uncertainty analysis).
2. **Analysis classifier:** Trained on outputs of RLCR with confidence scores (present within `<confidence>` tags) removed to prevent direct hacking.

As both RL models have comparable task accuracy, differences in classifier performance would indicate that the RLCR-trained model's reasoning chains contain information specifically useful for calibration. We train classifiers for 3 different model sizes of the Qwen-base model: 0.5B, 1.5B and 7B. Figure 7 shows Brier and ECE scores on HotPotQA-Modified. Interestingly, while 7B classifiers perform similarly, the *analysis classifier* outperforms the *baseline* at smaller sizes, suggesting classifier capacity is key. For a sufficiently expressive classifier (as with the 7B model), it is possible to infer confidence-relevant features directly from the solution. In contrast, smaller classifiers can make better use of RLCR reasoning chains. We believe that broader questions about the relationship between classifier capacity and CoT contents are an important topic for future work.

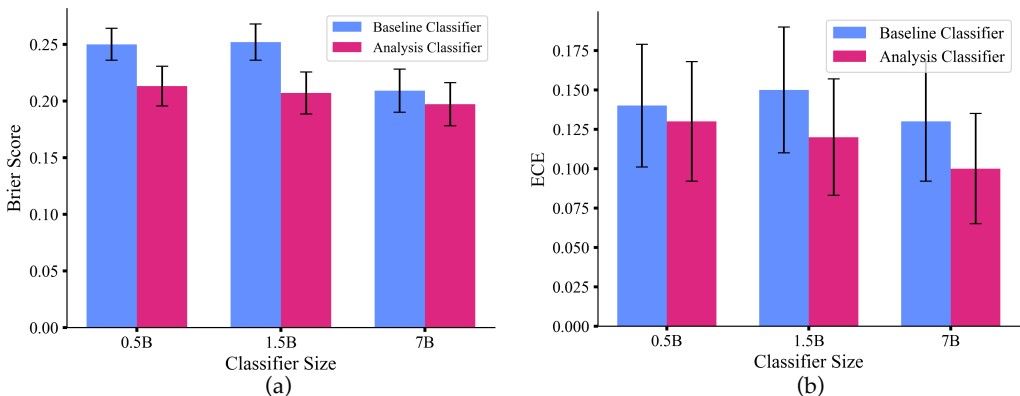

Figure 7: **Brier scores (a) and ECE (b) of baseline / analysis classifiers on HotPotQA-Modified across three model sizes.** Analysis classifiers outperform baselines at smaller sizes, suggesting that uncertainty CoT is essential for better calibration when capacity is limited.

# H  CONFIDENCE DISTRIBUTIONS

## H.1  CONFIDENCE DISTRIBUTIONS ACROSS DIFFERENT INPUTS

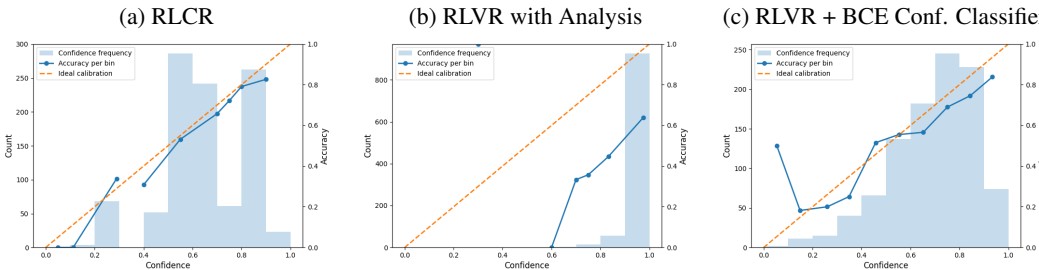

Figure 8: In-Distribution HotpotQA — calibration charts overlaid with confidence-frequency histograms for RLCR, RLVR, and RLVR+BCE Confidence Classifier.

Here, we present calibration charts overlaid with confidence–frequency histograms. For an ideally calibrated model, the accuracy-calibration curve would lie close to the orange dashed line. The histograms directly address whether a model's confidence values are genuinely diverse or instead clustered around a narrow range.

In Figure 8a, we show results for In Distribution evaluation for HotpotQA. For RLCR, the histogram shows that the model uses the full confidence range, with substantial mass in mid-confidence bins (0.4–0.8) and non-trivial usage of both lower and higher bins. This indicates that RLCR produces input-dependent confidence scores rather than collapsing toward a single accuracy-like value. The accuracy-per-bin curve also closely tracks the ideal diagonal, demonstrating that these varied confidence levels correspond to meaningful differences in correctness likelihood.

In contrast, RLVR (Figure 8b) concentrates almost all predictions in the 0.9–1.0 range, with very little representation of lower bins. This clustering suggests that RLVR's outputs are largely uniform and overconfident, and that its confidence does not meaningfully vary across inputs.

Finally, when we use an RLVR-trained model as the base for a BCE confidence classifier (Figure 8c), the accuracy-per-bin curve aligns more closely with the ideal line compared to the RLVR plots, suggesting that the classifier is also effective. However, it still underperforms RLCR.

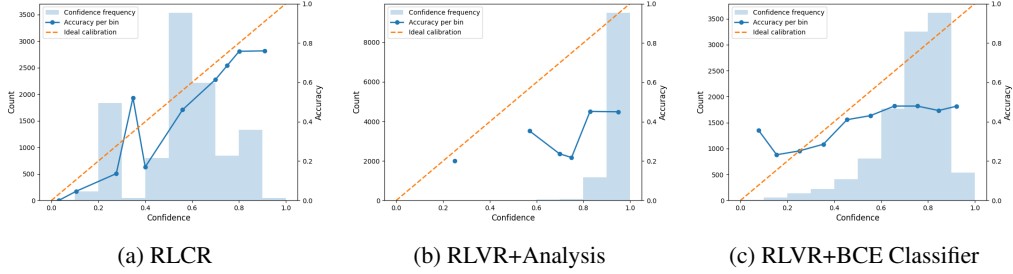

Figure 9: Out-of-Distribution averaged over 6 datasets — calibration charts overlaid with confidence-frequency histograms for RLCR, RLVR, and RLVR+BCE Confidence Classifier.

We observe a similar pattern in our out-of-distribution evaluations. Results are shown in Figure 9. RLVR (Figure 9b) continues to produce confidence values that collapse to 1.0 regardless of accuracy, and its accuracy-per-bin curve deviates substantially from ideal calibration—for example, predictions with verbalized confidences of 0.8–0.9 achieve only around 0.3 accuracy on average. Training a BCE confidence classifier on top of the RLVR model (Figure 9c) mitigates this behavior to some extent, yielding a *slightly* more distributed spread of confidence values and an accuracy-per-bin curve that moves a bit closer to the ideal line. However, RLCR (Figure 9a) still performs best: it exhibits a

healthy spread of confidence values across the full range (0–1) and an accuracy-per-bin curve that remains much closer to ideal calibration than either RLVR or its BCE classifier variant.

## H.2 PER INPUT CONFIDENCE DISTRIBUTION

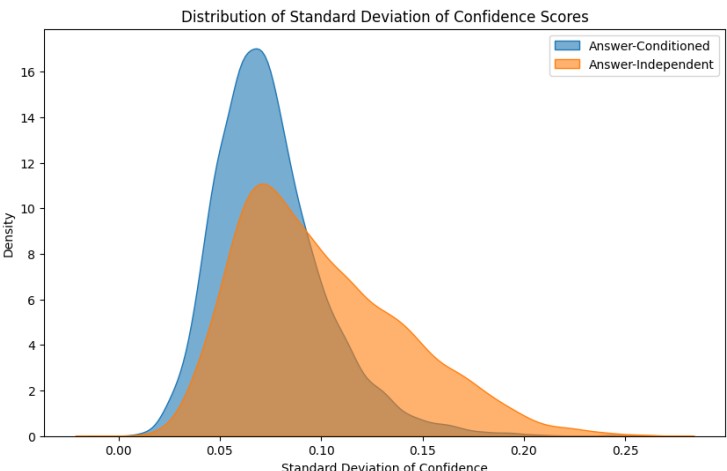

Figure 10: Distribution of Standard Deviation of Answer-Conditioned and Answer-Independent confidence scores

To analyze how the model's confidence values are distributed for a given question, we conduct an analysis of per-input confidence variability. For each question, we compute the standard deviation of the model's confidence scores across sampled generations, and the plot shows the distribution of these per-question standard deviations.

We plot two distributions:

1. **Answer-Conditioned Distribution:** $p(c|x, y, a)$. For a given question, solution, and final answer, we sample $N = 16$ analysis/CoT trajectories and compute the standard deviation of their confidence scores. This reflects how stable the model's confidence is for a given reasoning trajectory and answer. Intuitively, a high standard deviation here would be undesirable, as it would indicate that the model is internally inconsistent or "confused" about its confidence for a given solution.

2. **Answer-Independent Distribution:** $p(c|x)$. We sample N = 16 full CoTs, each with unique reasoning paths. These trajectories correspond to a variety of distinct reasoning paths (and may include multiple different answers). We then compute the standard deviation of the confidences over these trajectories for the same question, capturing how confidence varies across different solutions produced by the model for a particular question.

As shown in Figure 10, the answer-independent distribution (orange) has significantly more mass to the right, indicating noticeably higher variability in confidence across different reasoning trajectories. In contrast, the answer-conditioned distribution (blue) is narrower - reflecting reduced noise once the answer is fixed - but importantly, it still displays a non-trivial spread rather than collapsing near zero.

Together, these findings show that when generating multiple reasoning trajectories for the same question, the confidence scores have noticeably higher variance compared to generating multiple confidence scores for a fixed reasoning trajectory.

## I    COMPARISON TO RL-CALIBRATION BASELINES

| Method | HotpotQA | | | | O.O.D Averaged | | | |
|---|---|---|---|---|---|---|---|---|
| | Acc. | AUROC | Brier | ECE | Acc. | AUROC | Brier | ECE |
| Calibration RL (Full Seq) | 0.00 | N.A | **0.00** | **0.00** | 0.00 | N.A | **0.00** | **0.00** |
| Calibration RL (Analysis + Conf) | 62.0% | 0.53 | 0.24 | 0.08 | 52.8% | 0.57 | 0.27 | 0.25 |
| Abstention-RL | 62.1% | 0.61 | 0.32 | 0.31 | 54% | 0.59 | 0.35 | 0.35 |
| RLVR | **63.0**% | 0.50 | 0.37 | 0.37 | 53.9% | 0.50 | 0.46 | 0.46 |
| RLCR (**ours**) | 62.1% | **0.69** | 0.21 | 0.03 | **56.2%** | **0.68** | 0.21 | 0.21 |

Table 6: **Results of RL-calibration baselines**. All baselines significantly outperform RLVR on calibration, but underperform against RLCR on both accuracy and calibration. Training only with calibration rewards leads to degenerate solutions. This collapse can be avoided by masking loss over the the answer and think tokens. Training models to abstain never explicitly teaches confidence estimation and can also suppress exploration.

**Baselines.**    We compare RLCR against carefully adapted variants of the most relevant RL-for-calibration baselines. Several prior approaches focus primarily on calibration and do not explicitly optimize accuracy. To ensure fair comparison, we adapt these methods to the reasoning setting by initializing these baselines from our RLVR model, which has already been trained for accuracy.

1. **Calibration RL (Full-Sequence)** (Stangel et al., 2025; Xu et al., 2024): We train a variant that optimizes only the Brier-score reward, without any accuracy-dependent term. We apply the loss over the entire generation (`think`, `answer`, `analysis`, `confidence`). This represents the most direct application of a proper-scoring-rule reward to the reasoning-LLM setting. We initialize from the RLVR model.

2. **Calibration RL (Analysis+Confidence Only).** Prior work (Stangel et al., 2025) demonstrates that applying calibration rewards only to the analysis and confidence portions can stabilize training and preserve task accuracy. We therefore implement a stronger, accuracy-preserving variant of the above baseline by restricting the reward to the analysis and confidence spans while taking no loss over the thinking/answer. Similar to the above baseline, we initialize from the RLVR model.

3. **Abstention-RL** (Wei et al., 2025; Mohamadi et al., 2025; Song et al., 2025): Recent papers propose ternary rewards that give +1 for correct answers, 0 for incorrect answers, and an intermediate reward $\lambda$ for explicit abstentions (e.g., "I don't know"). We adopt this family of methods with $\lambda = 0.5$, a standard midpoint value. Because abstention models do not produce confidence scores, we evaluate calibration by using a test-time prompt that instructs the model to *never* abstain. As this method directly optimizes for accuracy as well, we initialize from the standard Qwen2.5-7B model.

**Results.**    Table 6 reports the full results. All baselines significantly outperform vanilla RLVR on calibration, but underperform against RLCR on all metrics.

*Calibration RL (Full-Sequence)* collapses to near-zero accuracy: with no reward shaping for correctness, the model rapidly converges to a degenerate but reward-maximizing behavior, outputting empty or trivial answers with confidence 0. This yields perfect calibration under the Brier score and is essentially the optimal policy with a calibration-only reward. While KL-regularization can potentially reduce this collapse, there is constant pressure to reduce task accuracy with this variant.

*Calibration RL (Analysis+Confidence Only)* prevents collapse and maintains accuracy comparable to RLCR, but calibration remains noticeably weaker. We hypothesize two contributing factors. First, the RLVR models might not be good starting points for further RL optimization. They might have reduced entropy or a very different output distribution compared to the base model. Second, jointly optimizing accuracy and calibration, as RLCR does, might provide complementary gradient signals that reinforce one another and enable more effective learning of confidence estimation.

*Abstention-RL* underperforms RLCR on calibration. Because the abstention reward only teaches whether the model's internal confidence exceeds the threshold $\lambda$, it never learns fine-grained confidence estimation. Moreover, rewarding abstention can also suppress exploration: once the model learns to abstain on difficult questions, it may no longer attempt them, limiting both reasoning improvement and calibration learning. Abstention rewards are better suited for settings where the primary goal is not to improve accuracy, but rather teach model the skill to abstain.

## J    ANALYZING SFT+RLCR

| Method | HotpotQA | | | | O.O.D Averaged | | | |
|---|---|---|---|---|---|---|---|---|
| | **Acc.** | **AUROC** | **Brier** | **ECE** | **Acc.** | **AUROC** | **Brier** | **ECE** |
| Base | 56.1% | 0.56 | 0.40 | 0.39 | 47.8% | 0.53 | 0.46 | 0.45 |
| RLVR | **72.9%** | 0.47 | 0.28 | 0.26 | **52.5%** | 0.52 | 0.49 | 0.49 |
| RLCR | 72.7% | 0.67 | 0.17 | 0.10 | 50.9% | 0.60 | 0.28 | 0.25 |
| SFT+RLCR (original) | 72.2% | **0.78** | **0.14** | **0.08** | 43.8% | **0.66** | **0.24** | **0.18** |
| SFT+RLCR (tweaked prompt) | 72.5% | 0.75 | 0.15 | 0.09 | 49.8% | 0.62 | 0.25 | 0.21 |

Table 7: **Results of SFT+RLCR with simple prompt change**. Adding a single line to the prompt boosts O.O.D accuracy from 43.8% to 48%.

The results in Table 1 showed that our SFT+RLCR model experienced a substantial drop in O.O.D. accuracy. While RLVR and vanilla RLCR achieve O.O.D. accuracies of 52.5% and 50.9% respectively, SFT+RLCR reaches only 43.8%. To investigate this degradation, we manually examined SFT+RLCR's generations and uncovered a consistent abnormality unique to this model: the model often identifies the correct answer during its reasoning, but then places an incorrect, seemingly random number inside the answer tags. This behavior suggests that extended RL training on Math-heavy data may induce overfitting or domain-specific bias in how the model formats its final answer.

To test whether this issue reflects catastrophic forgetting or a more superficial misalignment, we reran the evaluation with minor modifications to the prompt. We added a single clarifying instruction:

*"Be careful about what you put in the answer tags. Do not arbitrarily put numbers there if the question has nothing to do with Math."*

Remarkably, this simple change improves O.O.D. accuracy from 43.8% to 49.8%, as shown in Table 7. This indicates that the degradation is not solely due to catastrophic forgetting; rather, most of the failure arises from formatting biases learned during Math-focused RL training, which are straightforward to reverse.

We hypothesize that introducing a small amount of KL regularization, or training on a more diverse RL dataset beyond Math, would mitigate these effects. Prior work has also observed that SFT can induce more forgetting than RL (Shenfeld et al., 2025; Mukherjee et al., 2025), and the remaining performance gap may indeed reflect residual forgetting—but to a much lesser extent than we initially suspected.

## K    HotPot Training Results

Fig. 11 shows the training curves for RLCR and RLVR. Both the correctness and calibration reward for RLCR increase smoothly, indicating that the model is able to jointly improve accuracy and calibration. Notably, the completion lengths of our method gradually increase during training as uncertainty reasoning improves.

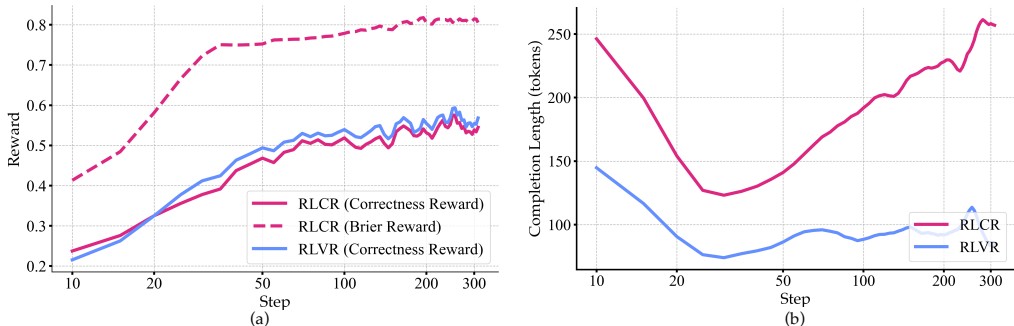

Figure 11: **(a) Reward curves for *RLCR* (ours) and *RLVR*.** Both correctness and calibration rewards improve under our method, demonstrating simultaneous gains in correctness and calibration. The Brier reward is shifted upward by 1 for clarity. **(b) Completion lengths during training.** The completion lengths of our method gradually increase during training as uncertainty reasoning improves.

## L    FULL RESULTS: MODELS TRAINED ON HOTPOTQA

**Discussion**    Below we present the full dataset-specific results. Although RLCR outperforms baselines on average, there can be considerable variance in calibration on individual O.O.D datasets. A particularly illustrative case is CommonsenseQA, where all methods achieve roughly 90% accuracy. RLVR is highly overconfident and consistently predicts 85–100% confidence across all questions and datasets. On CommonsenseQA this overconfidence happens to coincide with the dataset's high accuracy, producing deceptively strong calibration. Importantly, this alignment is a spurious correlation arising from RLVR's uniformly inflated confidence rather than genuine uncertainty modeling.

At the same time, we acknowledge that all methods, including RLCR, have significant room for improvement in O.O.D. calibration. We believe that extending RL training and incorporating a more diverse training dataset can further strengthen robustness and calibration.

| Method | SimpleQA | | | | Trivia | | | |
|---|---|---|---|---|---|---|---|---|
| | **Acc.** | **AUROC** | **Brier** | **ECE** | **Acc.** | **AUROC** | **Brier** | **ECE** |
| Base | **13.5% ± 1.0** | 0.50 ± 0.01 | 0.77 ± 0.01 | 0.81 | 57.8% ± 2.2 | 0.51 ± 0.01 | 0.38 ± 0.02 | 0.37 |
| RLVR | | 0.50 ± 0.00 | 0.88 ± 0.01 | 0.88 | | 0.50 ± 0.00 | 0.38 ± 0.02 | 0.38 |
| RLVR+BCE Classifier | | 0.48 ± 0.03 | 0.53 ± 0.01 | 0.64 | | 0.57 ± 0.03 | 0.26 ± 0.01 | 0.15 |
| RLVR+Brier Classifier | 12.4% ± 1.0 | **0.60 ± 0.03** | **0.11 ± 0.01** | **0.06** | **62.2% ± 2.1** | 0.57 ± 0.03 | 0.37 ± 0.01 | 0.37 |
| RLVR+Probe | | 0.51 ± 0.03 | 0.14 ± 0.01 | 0.12 | | 0.47 ± 0.03 | 0.43 ± 0.01 | 0.41 |
| Answer Probability | | 0.42 ± 0.03 | 0.83 ± 0.01 | 0.85 | | 0.50 ± 0.03 | 0.37 ± 0.02 | 0.36 |
| RLCR (**ours**) | 12.1% ± 1.0 | **0.60 ± 0.02** | 0.24 ± 0.01 | 0.34 | 60.8% ± 2.1 | **0.73 ± 0.03** | **0.20 ± 0.01** | **0.06** |

Table 8: Performance on SimpleQA and Trivia datasets. Values indicate the mean with error margins given as half-widths of the 95% bootstrap confidence intervals. Best values bolded.

| Method | CommonsenseQA | | | | GPQA | | | |
|---|---|---|---|---|---|---|---|---|
| | **Acc.** | **AUROC** | **Brier** | **ECE** | **Acc.** | **AUROC** | **Brier** | **ECE** |
| Base | 88.6% ± 1.8 | 0.62 ± 0.05 | 0.10 ± 0.01 | **0.00** | 39.5% ± 4.4 | 0.49 ± 0.05 | 0.50 ± 0.04 | 0.51 |
| RLVR | | 0.50 ± 0.00 | 0.09 ± 0.02 | 0.09 | | 0.50 ± 0.00 | 0.60 ± 0.05 | 0.60 |
| RLVR+BCE Classifier | | 0.65 ± 0.06 | 0.12 ± 0.01 | 0.18 | | 0.52 ± 0.05 | 0.28 ± 0.02 | **0.16** |
| RLVR+Brier Classifier | 90.8% ± 1.6 | 0.65 ± 0.05 | 0.26 ± 0.01 | 0.42 | 39.5% ± 4.6 | 0.52 ± 0.06 | 0.29 ± 0.03 | 0.21 |
| RLVR+Probe | | 0.50 ± 0.06 | 0.75 ± 0.01 | 0.81 | | 0.50 ± 0.05 | 0.33 ± 0.04 | 0.29 |
| Answer Prob | | 0.60 ± 0.06 | **0.08 ± 0.01** | 0.03 | | 0.53 ± 0.05 | 0.54 ± 0.04 | 0.54 |
| RLCR (**ours**) | **91.3% ± 1.6** | **0.73 ± 0.06** | 0.17 ± 0.01 | 0.30 | **41.5% ± 4.7** | **0.55 ± 0.05** | **0.27 ± 0.01** | **0.16** |

Table 9: Performance on CommonsenseQA and GPQA datasets. Values indicate the mean with error margins given as half-widths of the 95% bootstrap confidence intervals. Best values bolded.

| Method | MATH-500 | | | | GSM8K | | | |
|---|---|---|---|---|---|---|---|---|
| | **Acc.** | **AUROC** | **Brier** | **ECE** | **Acc.** | **AUROC** | **Brier** | **ECE** |
| Base | 46.8% ± 4.30 | 0.59 ± 0.05 | 0.48 ± 0.04 | 0.49 | 73.5% ± 2.50 | 0.52 ± 0.03 | 0.24 ± 0.02 | 0.22 |
| RLVR | | 0.50 ± 0.01 | 0.61 ± 0.04 | 0.61 | | 0.50 ± 0.00 | 0.20 ± 0.02 | 0.20 |
| RLVR + Classifier | | 0.70 ± 0.05 | 0.26 ± 0.02 | 0.22 | | 0.57 ± 0.04 | 0.16 ± 0.02 | **0.07** |
| RLVR + Brier Classifier | 38.6% ± 4.20 | 0.58 ± 0.05 | 0.32 ± 0.03 | 0.29 | 80.1% ± 2.10 | 0.66 ± 0.04 | 0.55 ± 0.02 | 0.63 |
| RLVR + Probe | | 0.67 ± 0.05 | **0.24 ± 0.03** | **0.15** | | 0.53 ± 0.04 | 0.41 ± 0.01 | 0.48 |
| Answer Probability | | **0.79 ± 0.05** | 0.51 ± 0.04 | 0.55 | | **0.78 ± 0.02** | 0.16 ± 0.02 | 0.17 |
| RLCR (**ours**) | **45.4% ± 4.50** | 0.72 ± 0.05 | 0.25 ± 0.02 | 0.19 | **86.3% ± 1.75** | 0.74 ± 0.04 | **0.14 ± 0.01** | 0.20 |

Table 10: Performance on Math-500 and GSM8K datasets. Values indicate the mean with error margins given as half-widths of the 95% bootstrap confidence intervals. Best values bolded.

| Method | HotpotQA | | | | HotpotQA-Modified | | | |
|---|---|---|---|---|---|---|---|---|
| | Acc. | AUROC | Brier | ECE | Acc. | AUROC | Brier | ECE |
| Base | 39.7% ± 3.10 | 0.54 ± 0.02 | 0.53 ± 0.03 | 0.53 | 30.4% ± 4.30 | 0.59 ± 0.04 | 0.57 ± 0.04 | 0.59 |
| RLVR | | 0.50 ± 0.00 | 0.37 ± 0.03 | 0.37 | | 0.50 ± 0.00 | 0.54 ± 0.05 | 0.54 |
| RLVR +BCE Classifier | | 0.66 ± 0.04 | 0.22 ± 0.01 | 0.07 | | 0.77 ± 0.05 | 0.21 ± 0.02 | 0.13 |
| RLVR +Brier Classifier | 63.0% ± 3.05 | 0.65 ± 0.04 | 0.22 ± 0.02 | 0.09 | 46.0% ± 4.30 | 0.79 ± 0.05 | 0.20 ± 0.02 | 0.12 |
| RLVR +Probe | | 0.55 ± 0.04 | 0.24 ± 0.01 | 0.10 | | 0.57 ± 0.05 | 0.26 ± 0.01 | 0.12 |
| Answer Prob | | **0.72 ± 0.04** | 0.36 ± 0.03 | 0.36 | | 0.61 ± 0.05 | 0.52 ± 0.04 | 0.53 |
| RLCR (**ours**) | 62.1% ± 3.05 | 0.69 ± 0.04 | **0.21 ± 0.01** | **0.03** | 44.4% ± 4.20 | **0.80 ± 0.05** | **0.19 ± 0.02** | **0.08** |

Table 11: Performance on HotpotQA and HotpotQA-Modified datasets. Values indicate the mean with error margins given as half-widths of the 95% bootstrap confidence intervals. Best values bolded.

## M  FULL RESULTS: MODELS TRAINED ON MATH

| Method | SimpleQA | | | | TriviaQA | | | |
|---|---|---|---|---|---|---|---|---|
| | Acc. | AUROC | Brier | ECE | Acc. | AUROC | Brier | ECE |
| Base | 13.5% ± 1.01 | 0.50 ± 0.01 | 0.77 ± 0.01 | 0.81 | 57.8% ± 2.25 | 0.51 ± 0.01 | 0.38 ± 0.02 | 0.37 |
| RLVR | | 0.53 ± 0.02 | 0.83 ± 0.01 | 0.84 | | 0.50 ± 0.02 | 0.43 ± 0.02 | 0.43 |
| RLVR+BCE Classifier | | 0.45 ± 0.02 | 0.57 ± 0.01 | 0.64 | | 0.57 ± 0.03 | 0.29 ± 0.01 | 0.22 |
| RLVR+Brier Classifier | **15.2% ± 1.05** | 0.49 ± 0.02 | **0.15 ± 0.01** | **0.11** | 58.3% ± 2.20 | 0.61 ± 0.03 | 0.30 ± 0.01 | 0.25 |
| RLVR+Probe | | 0.44 ± 0.02 | 0.58 ± 0.01 | 0.66 | | 0.56 ± 0.03 | 0.30 ± 0.01 | 0.23 |
| Answer Prob | | 0.45 ± 0.02 | 0.80 ± 0.01 | 0.81 | | 0.48 ± 0.02 | 0.40 ± 0.02 | 0.38 |
| RLCR (**ours**) | 12.0% ± 0.95 | 0.52 ± 0.02 | 0.43 ± 0.01 | 0.54 | **61.0% ± 2.13** | 0.67 ± 0.02 | 0.22 ± 0.01 | 0.10 |
| RLCR+SFT (**ours**) | 11.4% ± 0.94 | **0.60 ± 0.03** | 0.29 ± 0.01 | 0.40 | 55.6% ± 2.20 | **0.72 ± 0.02** | **0.21 ± 0.01** | **0.06** |

Table 12: Performance on SimpleQA and TriviaQA datasets. Values indicate the mean with error margins given as half-widths of the 95% bootstrap confidence intervals. Best values bolded.

| Method | CommonsenseQA | | | | GPQA | | | |
|---|---|---|---|---|---|---|---|---|
| | Acc. | AUROC | Brier | ECE | Acc. | AUROC | Brier | ECE |
| Base | 88.6% ± 1.76 | 0.62 ± 0.05 | **0.10 ± 0.01** | **0.00** | 39.5% ± 4.35 | 0.49 ± 0.05 | 0.50 ± 0.04 | 0.51 |
| RLVR | | 0.55 ± 0.02 | 0.13 ± 0.02 | 0.13 | | 0.50 ± 0.02 | 0.53 ± 0.04 | 0.53 |
| RLVR+BCE Classifier | | 0.61 ± 0.05 | 0.23 ± 0.01 | 0.34 | | 0.51 ± 0.05 | 0.33 ± 0.03 | 0.27 |
| RLVR+Brier Classifier | 89.3% ± 1.88 | 0.60 ± 0.05 | 0.30 ± 0.01 | 0.45 | **50.0% ± 4.91** | 0.53 ± 0.05 | 0.33 ± 0.03 | 0.28 |
| RLVR+Probe | | 0.57 ± 0.05 | 0.19 ± 0.01 | 0.28 | | 0.50 ± 0.05 | 0.30 ± 0.02 | 0.19 |
| Answer Prob | | 0.56 ± 0.05 | 0.10 ± 0.02 | 0.09 | | 0.53 ± 0.06 | 0.44 ± 0.04 | 0.40 |
| RLCR (**ours**) | **90.1% ± 1.68** | 0.62 ± 0.05 | 0.21 ± 0.01 | 0.34 | 43.3% ± 4.46 | 0.57 ± 0.05 | 0.26 ± 0.02 | 0.10 |
| SFT+RLCR (**ours**) | 77.6% ± 2.17 | **0.73 ± 0.04** | 0.22 ± 0.02 | 0.25 | 32.6% ± 4.35 | **0.60 ± 0.06** | **0.23 ± 0.02** | **0.08** |

Table 13: Performance on CommonsenseQA and GPQA datasets. Values indicate the mean with error margins given as half-widths of the 95% bootstrap confidence intervals. Best values bolded.

| Method | MATH-500 | | | | GSM8K | | | |
|---|---|---|---|---|---|---|---|---|
| | Acc. | AUROC | Brier | ECE | Acc. | AUROC | Brier | ECE |
| Base | $46.8\% \pm 4.20$ | $0.59 \pm 0.05$ | $0.48 \pm 0.04$ | 0.49 | $73.5\% \pm 2.46$ | $0.52 \pm 0.03$ | $0.24 \pm 0.02$ | 0.22 |
| RLVR | | $0.45 \pm 0.04$ | $0.44 \pm 0.04$ | 0.43 | | $0.47 \pm 0.05$ | $0.09 \pm 0.01$ | **0.05** |
| RLVR +BCE Classifier | | $0.77 \pm 0.05$ | $0.22 \pm 0.03$ | 0.18 | | $\mathbf{0.77 \pm 0.05}$ | $\mathbf{0.08 \pm 0.01}$ | 0.06 |
| RLVR +Brier Classifier | $59.2\% \pm 4.10$ | $0.79 \pm 0.05$ | $0.22 \pm 0.03$ | 0.18 | $\mathbf{90.6\% \pm 1.57}$ | $0.76 \pm 0.05$ | $\mathbf{0.08 \pm 0.01}$ | **0.05** |
| RLVR +Probe | | $0.67 \pm 0.05$ | $0.26 \pm 0.03$ | 0.20 | | $0.56 \pm 0.05$ | $0.11 \pm 0.01$ | 0.14 |
| Answer Prob | | $0.54 \pm 0.05$ | $0.39 \pm 0.04$ | 0.39 | | $0.48 \pm 0.02$ | $0.09 \pm 0.02$ | 0.09 |
| RLCR (**ours**) | $\mathbf{59.8\% \pm 4.20}$ | $0.67 \pm 0.04$ | $0.23 \pm 0.02$ | **0.12** | $89.6\% \pm 1.71$ | $0.63 \pm 0.03$ | $0.10 \pm 0.01$ | 0.12 |
| SFT+RLCR (**ours**) | $55.8\% \pm 4.40$ | $\mathbf{0.81 \pm 0.05}$ | $\mathbf{0.19 \pm 0.03}$ | 0.16 | $90.4\% \pm 1.67$ | $0.73 \pm 0.04$ | $\mathbf{0.08 \pm 0.01}$ | 0.06 |

Table 14: Performance on MATH-500 and GSM8K datasets. Values indicate the mean with error margins given as half-widths of the 95% bootstrap confidence intervals. Best values bolded.

| Method | Big-Math Digits | | | | HotpotQA | | | |
|---|---|---|---|---|---|---|---|---|
| | Acc. | AUROC | Brier | ECE | Acc. | AUROC | Brier | ECE |
| Base | $48.1\% \pm 3.10$ | $0.56 \pm 0.03$ | $0.47 \pm 0.03$ | 0.47 | $39.7\% \pm 3.10$ | $0.54 \pm 0.02$ | $0.53 \pm 0.03$ | 0.53 |
| RLVR | | $0.50 \pm 0.03$ | $0.32 \pm 0.03$ | 0.30 | | $0.50 \pm 0.01$ | $0.55 \pm 0.03$ | 0.55 |
| RLVR+BCE Classifier | | $\mathbf{0.81 \pm 0.04}$ | $0.16 \pm 0.02$ | 0.05 | | $0.58 \pm 0.03$ | $0.28 \pm 0.01$ | 0.20 |
| RLVR+Brier-Classifier | $68.8\% \pm 2.90$ | $0.79 \pm 0.04$ | $0.17 \pm 0.02$ | 0.05 | $\mathbf{49.7\% \pm 3.15}$ | $0.60 \pm 0.04$ | $0.30 \pm 0.02$ | 0.24 |
| RLVR+Probe | | $0.72 \pm 0.04$ | $0.19 \pm 0.02$ | 0.07 | | $0.58 \pm 0.04$ | $0.28 \pm 0.01$ | 0.16 |
| Answer Prob | | $0.53 \pm 0.04$ | $0.30 \pm 0.03$ | 0.30 | | $0.60 \pm 0.03$ | $0.47 \pm 0.03$ | 0.47 |
| RLCR (**ours**) | $68.7\% \pm 2.75$ | $0.73 \pm 0.03$ | $0.18 \pm 0.01$ | 0.05 | $48.3\% \pm 3.00$ | $0.61 \pm 0.04$ | $0.27 \pm 0.01$ | 0.18 |
| RLCR+SFT (**ours**) | $\mathbf{70.5\% \pm 2.70}$ | $0.79 \pm 0.04$ | $\mathbf{0.15 \pm 0.02}$ | **0.03** | $41.6\% \pm 3.15$ | $\mathbf{0.65 \pm 0.04}$ | $\mathbf{0.24 \pm 0.01}$ | **0.09** |

Table 15: Performance on Big-Math Digits and HotpotQA. Values indicate the mean with error margins given as half-widths of the 95% bootstrap confidence intervals. Best values bolded.

## SIMPLEQA EXAMPLE

---

**Question**

Who was awarded the Oceanography Society's Jerlov Award in 2018? (Ground Truth Answer: Annick Bricaud)

---

**RLCR (ours)**

<think> The question asks who was awarded the Oceanography Society's Jerlov Award in 2018. The Jerlov Award is given by the Oceanography Society, and it is named after the late Professor Gunnar Jerlov, a pioneer in the field of ocean optics. To find the winner of the 2018 award, I would need to look up the most recent information on the Oceanography Society's website or other reliable sources. </think>

<answer> Dr. David W. Schiedek </answer>

<analysis> The Oceanography Society's website or other reliable sources do not provide information on the winner of the 2018 Jerlov Award. It is possible that the information is not readily available or that the award was not given in 2018. Additionally, the name "Dr. David W. Schiedek" is not a well-known figure in the field of oceanography, which raises uncertainty about the accuracy of this answer. </analysis>

<confidence> 0.25 </confidence>

---

**RLVR**

<think> The question asks for the person who was awarded the Oceanography Society's Jerlov Award in 2018. The Jerlov Award is given by the Oceanography Society to recognize outstanding contributions to the field of oceanography. The recipient for 2018 was Dr. Laurence C. Smith. </think>

<answer> Laurence C. Smith </answer>

<confidence> 100. </confidence>

---

**Qwen-7B-Base**

<think> To find out who was awarded the Oceanography Society's Jerlov Award in 2018, I will search for the most recent information on the award winners from the Oceanography Society's official website or a reliable source that covers scientific awards. </think>

<answer> The Oceanography Society's Jerlov Award in 2018 was awarded to Dr. Lisa Levin. </answer>

<confidence> 0.95 </confidence>

---

HOTPOTQA-MODIFIED EXAMPLE

All supporting information from the given question has been removed. The only way to answer correctly is if the model remembers the information in its weights.

---

**Question and Supporting Information**

**Question:** Jacques Sernas, actor in *Fugitive in Trieste*, was of what nationality? (Ground Truth Answer: Lithuanian-born French)
Your answer will be verified with exact match score. To ensure correct verification, only provide the answer within the <answer> </answer> tags. Do not put any sentences or reasoning process within the <answer> </answer> tags.

**Supporting Information:**
**Paragraph 0**
*Man From 1997* is a time travel episode of the 1956–57 anthology television series *Conflict* directed by Roy del Ruth, produced by Roy Huggins, written by James Gunn from a story by Alfred Bester, and starring Jacques Sernas, Charles Ruggles, Gloria Talbott and James Garner. The music was written by David Buttolph and the cinematographer was Ted D. McCord. The show was originally telecast on November 27, 1956 and a kinescope of the broadcast currently exists.

**Paragraph 1**
*Altair* is a 1956 Italian romantic drama film directed by Leonardo De Mitri and starring Franco Interlenghi, Antonella Lualdi and Jacques Sernas.

**Paragraph 2**
*Dieci canzoni d'amore da salvare* (English title: "Ten Love Songs") is a 1953 Italian film directed by Flavio Calzavara. The plot concerns a songwriter, played by Jacques Sernas, who leaves his sweetheart and publisher when he learns that he is going blind. Supporting Sernas were Brunella Bovo, Franca Tamantini, and Enrico Viarisio.

**Paragraph 3**
*Loving You Is My Sin* (Italian: *Amarti è il mio peccato*) is a 1953 Italian melodrama film directed by Sergio Grieco and starring Jacques Sernas, Luisa Rossi and Elisa Cegani.

**Paragraph 4**
*It's All Adam's Fault* or *In Six Easy Lessons* (French: *C'est la faute d'Adam*) is a 1958 French comedy film directed by Jacqueline Audry and starring Dany Robin, Jacques Sernas and Mijanou Bardot.

**Paragraph 5**
*Balearic Caper* (Spanish: *"Zarabanda bing bing"*, Italian: *"Baleari Operazione Oro"*, French: *"Barbouze chérie"*, also known as *Operation Gold*) is a 1966 Spanish-Italian-French heist-Eurospy comedy film written and directed by José María Forqué and starring Jacques Sernas, Daniela Bianchi and Mireille Darc. It was shot in Ibiza.

**Paragraph 6**
*Finishing School* (Italian: *Fanciulle di lusso*) is a 1953 French-Italian comedy film directed by Bernard Vorhaus and starring Susan Stephen, Anna Maria Ferrero and Jacques Sernas. It was made at Cinecittà with sets designed by the art director Franco Lolli. It is also known by the alternative title of *Luxury Girls*.

**Paragraph 7**
*Jump into Hell* is a 1955 war film directed by David Butler. The film stars Jacques Sernas (billed as "Jack Sernas") and Kurt Kasznar. The first contemporary Hollywood war film of the war in Indochina, the story is a fictionalized account of the Battle of Dien Bien Phu.

**RLCR**

`<think>` The question asks for the nationality of Jacques Sernas, the actor in Fugitive in Trieste. The information provided mentions Jacques Sernas in several films, but does not specify his nationality. However, it does mention that he starred in Altair, a 1956 Italian romantic drama film, and that he was an actor in several French and Italian films. This suggests that Jacques Sernas might be of French or Italian nationality, as he starred in films produced in those countries. However, there is no direct information about his nationality in the given text. `</think>`

`<answer>` French `</answer>`

`<analysis>` The information provided mentions Jacques Sernas in several French and Italian films, and he starred in Altair, a 1956 Italian romantic drama film. However, there is no direct information about his nationality. While it is possible that he could be of French or Italian nationality, the lack of specific information makes it uncertain. Additionally, the fact that he starred in both French and Italian films could suggest that he might be of mixed nationality or have connections to both countries. Due to the ambiguity and lack of direct information, the confidence in this answer is relatively low. `</analysis>`

`<confidence>` 0.4 `</confidence>`

**RLVR**

`<think>` The question asks for the nationality of Jacques Sernas, the actor in Fugitive in Trieste. Jacques Sernas was an American actor. `</think>`

`<answer>` American `</answer>`

**Qwen-7B-Base**

`<think>` To determine Jacques Sernas' nationality, I will analyze the information provided in the supporting information.
From the information given, I can see that Jacques Sernas has appeared in various films from different countries, such as Italy, France, and Spain. However, there is no specific mention of his nationality in the provided information.
Since there is no direct information about Jacques Sernas' nationality, I will analyze the context of the question. The question is about Jacques Sernas' nationality in the context of the film "Fugitive in Trieste." Trieste is a city in Italy, and the film is an Italian romantic drama.
Based on this context, it is reasonable to assume that Jacques Sernas is of Italian nationality. However, without any direct information, I cannot be 100% confident in this assumption. `</think>`

`<answer>` Italian `</answer>`

`<confidence>` 0.8 `</confidence>`

## MATH EXAMPLE

An example where the analysis from the vanilla RLCR is generic, while the SFT+RLCR model produces a much more specific evaluation that directly references the reasoning process. Although the RLCR model achieves reasonable calibration on Math, the SFT+RLCR model outperforms it, likely because it has learned to generate more effective uncertainty-aware chain-of-thought reasoning.

---

**Question**

Dalton, Hunter, Alex, Bella, and Chris formed a Superhero Fan Club. They set a goal to watch as many Superhero movies as they could in one summer. Dalton watched 15 movies, Hunter watched 19, Alex watched 25, Bella watched 21, and Chris watched 11. They all watched 5 superhero movies together, while Dalton, Hunter, and Alex watched an additional 3 movies only among themselves, and Bella and Chris watched 2 more movies only with each other. Every other movie they watched was different. How many different movies did they see? (Ground Truth Ans: 63)

---

**RLCR (ours)**

`<think>` To determine the total number of different movies watched by Dalton, Hunter, Alex, Bella, and Chris, we need to account for the movies they watched together and the additional movies they watched among themselves or with each other.

First, let's calculate the total number of movies watched if we didn't account for any overlaps: - Dalton watched 15 movies. - Hunter watched 19 movies. - Alex watched 25 movies. - Bella watched 21 movies. - Chris watched 11 movies.

The sum of these is:

$$15 + 19 + 25 + 21 + 11 = 91$$

Next, we need to subtract the overlaps. They all watched 5 movies together, so we subtract 4 overlaps (since the 5 movies are counted 5 times in the initial sum):

$$91 - 4 \times 5 = 91 - 20 = 71$$

Now, we need to account for the additional movies watched among specific groups: - Dalton, Hunter, and Alex watched 3 additional movies only among themselves. These movies are already subtracted in the previous step, so we don't need to subtract them again. - Bella and Chris watched 2 additional movies only with each other. These movies are also already subtracted in the previous step, so we don't need to subtract them again.

Since all the overlaps have been accounted, the total number of different movies they saw is 71.

`<answer>` 71 `</answer>`

`<analysis>` The solution provided involves a reasonable approach to accounting for the overlaps in the number of movies watched. However, there is uncertainty in the interpretation of the problem statement, particularly the part about "Every other movie they watched was different."

This phrase could be interpreted in two ways: 1. That the 5 movies watched together, the 3 additional movies watched by Dalton, Hunter, and Alex, and the 2 additional movies watched by Bella and Chris are all distinct from each other and from the other movies watched by the individuals. 2. That the total number of movies watched by each individual, minus the overlaps, results in a set of distinct movies.

Given the information provided, the first interpretation seems more plausible, but it introduces an element of uncertainty. The problem does not clearly state that all movies watched are distinct, only that "Every other movie they watched was different," which could be interpreted as a subset of the total movies watched.

Therefore, the confidence level in the solution is set at 0.6, acknowledging the potential for misinterpretation and the lack of clear information in the problem statement.`</analysis>`

`<confidence>` 0.6 `</confidence>`

---

**SFT+RLCR (ours)**

<think> To determine the total number of different movies the Superhero Fan Club members watched, we need to account for the movies watched individually and in groups, ensuring we don't double-count any movies.

First, let's sum the total number of movies watched by each member without considering overlaps: - Dalton watched 15 movies. - Hunter watched 19 movies. - Alex watched 25 movies. - Bella watched 21 movies. - Chris watched 11 movies.

Adding these gives:

$$15 + 19 + 25 + 21 + 11 = 91$$

Next, we need to subtract the overlaps. They all watched 5 movies together, so we subtract 4 times 5 (since the 5 movies are counted 5 times in the initial sum, once for each person):

$$91 - 4 \times 5 = 91 - 20 = 71$$

Now, we need to account for the additional movies watched in smaller groups. Dalton, Hunter, and Alex watched 3 more movies only among themselves. These 3 movies are already included in the initial counts for Dalton, Hunter, and Alex, but we need to ensure they are not double-counted in the all-together group. Since the 3 movies are only among these three, they are not part of the 5 movies watched together, so we don't need to subtract them again. They are already included in the individual counts and the group of 5, so we add them back to the total after the initial subtraction:

$$71 + 3 = 74$$

Bella and Chris watched 2 more movies only with each other. These 2 movies are already included in the initial counts for Bella and Chris, but we need to ensure they are not double-counted. Since the 2 movies are only between Bella and Chris, they are not part of the 5 movies watched together or the 3 movies watched by Dalton, Hunter, and Alex, so we add them back to the total after the initial subtraction:

$$74 + 2 = 76$$

At this point, we have accounted for all the overlaps, and the total number of different movies they saw is 76. To verify, let's consider the structure of the problem: - The 5 movies watched together are counted 5 times initially but subtracted 4 times, so they are correct. - The 3 movies watched by Dalton, Hunter, and Alex are included in their individual counts and the group of 5, so they are correct. - The 2 movies watched by Bella and Chris are included in their individual counts, so they are correct.

The final count of 76 different movies is consistent with the problem constraints. Therefore, the total number of different movies they saw is 76. </think>

<answer> 76 </answer>

<analysis> Upon reviewing the solution, I notice that the reasoning process involves summing the total movies watched individually, subtracting the overlaps, and then adding back the group-specific movies. The initial sum of 91 is correct, and the subtraction of 20 for the 5 movies watched together is appropriate, resulting in 71. However, the addition of the 3 movies watched by Dalton, Hunter, and Alex (3) and the 2 movies watched by Bella and Chris (2) is problematic. These movies are already included in the individual counts and the group of 5, so adding them back is redundant and unnecessary. The reasoning does not clearly explain why these movies are added back, and the final total of 76 does not seem to be derived from a clear, step-by-step logical flow. The explanation about the 3 movies being "only among these three" and not part of the 5 is confusing, as the 3 movies are part of the individual counts and the group of 5, so they should not be added back. The 2 movies watched by Bella and Chris are also problematic, as they are only between Bella and Chris, but the reasoning does not clearly address how they are added back without double-counting. The final answer of 76 is not supported by the reasoning, and the steps are convoluted, making the solution unclear and potentially incorrect.</analysis>

<confidence> 0.3 </confidence>

---

**RLVR**

`<think>` To determine the total number of different movies watched by Dalton, Hunter, Alex, Bella, and Chris, we need to carefully account for the movies they watched together and the movies they watched exclusively among themselves and with each other.

1. **Total movies watched by each individual:** - Dalton: 15 movies - Hunter: 19 movies - Alex: 25 movies - Bella: 21 movies - Chris: 11 movies

2. **Movies watched together:** They all watched 5 movies together.

3. **Movies watched exclusively among themselves:** - Dalton, Hunter, and Alex watched an additional 3 movies only among themselves. - Bella and Chris watched 2 more movies only with each other.

4. **Calculating the total number of movies watched:** - First, sum the total movies watched by each individual:

$$15 + 19 + 25 + 21 + 11 = 91$$

- Subtract the movies watched together (since they are counted multiple times in the individual totals):

$$91 - 5 = 86$$

- Add back the movies watched together (they should only be counted once):

$$86 + 5 = 91$$

- Subtract the movies watched exclusively among themselves and with each other (since they are also counted multiple times): - Dalton, Hunter, and Alex watching 3 movies only among themselves are already subtracted when we subtracted the 5 movies watched together. - Bella and Chris watching 2 movies only with each other are already subtracted when we subtracted the 5 movies watched together. - Therefore, we do not need to make any further adjustments for these exclusive movies as they are already accounted for in the initial sum and subtraction.

5. **Final calculation:** The total number of different movies they saw is the sum of all individual movies watched, minus the movies watched together (to avoid double-counting), plus the movies watched together again (to include them once), and no further adjustments are needed for the exclusive movies:

$$91 - 5 + 5 = 91$$

Thus, the total number of different movies they saw is:

$$\boxed{91}$$

`</think>`

`<answer>` 91 `</answer>`

`<confidence>` 90 `</confidence>`

