# OpenReview forum: "Beyond Binary Rewards: Training LMs to Reason About Their Uncertainty"
_ICLR.cc/2026/Conference — ICLR 2026 Poster_

### Official Review · Reviewer_bCz3 · 2025-10-30

**Soundness:** 3
**Presentation:** 3
**Contribution:** 2
**Rating:** 4
**Confidence:** 4

**Summary:**

This paper introduces RLCR (Reinforcement Learning with Calibration Rewards), which augments standard binary-reward RL for reasoning LMs with a Brier score term to encourage calibrated confidence. Unlike conventional RL that promotes overconfidence, RLCR jointly optimizes accuracy and uncertainty estimation. The authors also provide some theoretical results showing that RLCR does not induce the model to generate uncertain and wrong answers. Experiments with Qwen2.5-7B on reasoning benchmarks show that RLCR maintains accuracy while improving calibration.

**Strengths:**

Strengths

- The paper is well written and easy to follow, with clear organization and motivation.

- Although using proper scoring rules for calibration is a well-established idea, the work’s contribution lies in proposing an RL objective that maintains accuracy while improving calibration in reasoning LMs, which is still an important direction to explore.

- The authors provide comprehensive empirical validation, showing that the proposed method consistently yields better-calibrated reasoning models without degrading accuracy across diverse tasks.

**Weaknesses:**

Weaknesses and Questions

- I would like to see a deeper analysis of the confidence distribution. For instance, are the confidence values truly diverse across different inputs, or do they mostly cluster around a narrow range regardless of the question type? Understanding this would clarify whether the observed calibration improvements reflect genuine uncertainty modeling or simply uniform confidence outputs near accuracy.

- While the paper theoretically explains why adding NLL instead of the Brier score breaks the correctness incentive, it would be informative to include an empirical comparison showing whether this theoretical difference actually affects performance or calibration in practice by conducting experiments with objectives that include NLL.

- The reported AUROC scores (around 0.5–0.6) seem relatively low; it’s unclear whether such values reflect meaningful discrimination ability. Some qualitative analysis or normalized baselines could help interpret these numbers more convincingly.

- It would be useful to examine how RLCR behaves in larger-scale or instruction-following settings, beyond the QA and math reasoning benchmarks. This would test whether the proposed calibration benefit generalizes to more complex reasoning domains.

- I’m also curious how the model behaves when generating multiple samples for the same question—does it assign noticeably different confidence scores to different (possibly contradictory) answers, or are the confidences largely similar across samples? This would give insight into how well the model captures epistemic uncertainty across reasoning trajectories.

- In Figure 4(a), the standard deviation of confidence across reasoning chains appears to be roughly 0.05–0.1, meaning that on a 0–100 scale, confidence typically fluctuates by 5–10 points, or up to 10–20 points across extremes. Doesn’t this suggest that the model’s confidence is still quite unstable across samples, rather than genuinely consistent? Some further interpretation of this behavior would help clarify the robustness of the claimed self-consistency.

**Questions:**

See above section

---

> ### Author Response · Authors · 2025-11-22
> **Response to Reviewer bCz3 (Part 1)**
>
> Thank you for your thoughtful review. We address your comments below.
>
> ---
>
> **Q: Clarifying Confidence Distribution Diversity**
>
> *Ans:* Thank you for the question - this is an important point. To further analyze the confidence distribution, we generated calibration charts overlaid with confidence-frequency histograms in **Figures 8 and 9 of Appendix H.1**. We create these plots for RLVR, RLCR and RLVR+Classifier, both in-distribution and O.O.D.
>
>
>
> These plots directly address whether the model’s confidence values are genuinely diverse or simply clustered around a narrow range. *RLCR exhibits a healthy spread of confidence values and an accuracy-per-bin curve that remains much closer to ideal calibration than either RLVR or its BCE classifier variant.*
>
>
>
> In contrast to RLCR, RLVR collapses its confidence predictions close to 1.0 and shows poor calibration. Adding a BCE confidence classifier on top of RLVR helps slightly by spreading out the confidence values and improving the accuracy-per-bin curve, but still remains less calibrated than RLCR.
>
>
>
> In our response to a later point below, we also analyze the distribution of confidence scores for a single question in **Figure 10 of Appendix H.2.**
>
>  ---
>
> **Q: Further Exploration into a possible NLL Objective**
>
> *Ans:* Thank you for the suggestion. We address this in a new **Appendix E** that: (a) provides an intuitive explanation of the differences between log and Brier scoring rules, (b) presents a toy experiment demonstrating that combining correctness with log score can lead to reward hacking while Brier does not, and (c) reports results on HotpotQA showing that log score marginally underperforms Brier score, although we do not observe hacking in this setting.
>
>
>
> **These experiments are discussed in detail in point (2) of the Global Response and in Appendix E of the updated paper.**
>
>
> ---
> **Q: AUROC Scores**
>
> *Ans:* Our AUROC scores exceed 0.7 on in-distribution evaluations and fall in the high 0.6 range on OOD settings. While these OOD values indicate room for improvement, they still reflect non-trivial discrimination ability. We agree that stronger OOD performance may be achievable with larger-scale RL or training on a more diverse data distribution. To provide additional interpretability beyond AUROC, we also report ECE, which directly measures calibration quality. In the revised paper, we have included calibration plots that visually illustrate how RLCR achieves better-calibrated confidence estimates than baseline methods. We are happy to incorporate any additional metrics the reviewers believe would further strengthen this analysis.
>
>   ---
>
> **Q: Understanding Confidence Diversity for Multiple Generations of a Question**
>
> *Ans:* Thank you for the suggestion. We provide an analysis of per-input confidence variability in **Fig. 10 of Appendix H.2**. For each question, we compute the standard deviation of the model’s confidence scores across sampled generations, and the plot shows the following 2 distributions of these per-question standard deviations:
>
>
>
> (1) **Answer-Conditioned Distribution: $p(c | x, y, a)$**.
>
> For a given question $x$, solution $y$, and final answer $a$, we sample $N = 16$ uncertainty/confidence analysis trajectories and compute the standard deviation of their confidence scores. This reflects how stable the model’s confidence is for a given reasoning trajectory and answer. Intuitively, a high standard deviation here would be undesirable, as it would indicate that the model is internally inconsistent or “confused” about its confidence for the same given solution/answer.
>
>
>
> (2) **Answer-Independent Distribution: $p(c | x)$.**
>
> We sample $N = 16$ full CoTs, each with unique reasoning paths. These trajectories correspond to a variety of distinct reasoning paths (and may include multiple different answers). We then compute the standard deviation of the confidences over these trajectories for the same question, capturing how confidence varies across different solution trajectories.
>
>
>
> As shown in **Figure 10**, the answer-independent distribution (orange) has significantly more mass to the right, indicating noticeably higher variability in confidence across different reasoning trajectories. In contrast, the answer-conditioned distribution (blue) is narrower - reflecting reduced noise once the answer is fixed.
>
>
>
> Together, these findings show that when generating multiple reasoning trajectories for the same question, the confidence scores have noticeably higher variance compared to generating multiple confidence scores for a fixed reasoning trajectory.
>
>
>
>
>   ---
> (continued below)

---

> > ### Author Response · Authors · 2025-11-22
> > **Response to Reviewer bCz3 (Part 2)**
> >
> > (continued from above)
> >
> > ---
> >
> > **Q: SD of Confidence Across Reasoning Chains**
> >
> > *Ans:* We found that the Qwen-2.5-7B model converges to outputting confidence scores in multiples of 10 points (10,20,30,…). This naturally means that confidence can fluctuate by 5-10 points because the learned scale is not fine enough. We believe that larger models are capable of learning much finer scales, we have empirically observed GPT5 using a scale of 0.01 (1 point).
> >
> >
> >
> > We think the answer to this ultimately comes down to what the tolerable range for consistency is. For applications like factual QA and Math, 10 points might be tolerable, in contrast to critical applications like medicine where the acceptable tolerance might be much lower. We believe this consistency would also improve with larger models/longer training runs.
> >
> > ---
> > Thank you again for your review. We sincerely hope these clarifications and updates address your concerns and further strengthen your confidence in the work.

---

### Official Review · Reviewer_xB9Y · 2025-10-31

**Soundness:** 3
**Presentation:** 3
**Contribution:** 3
**Rating:** 6
**Confidence:** 4

**Summary:**

This paper proposes Reinforcement Learning with Calibration Rewards (RLCR), which improves the reward function of RLVR by combining with a term based on the Brier score over verbalized confidence estimates through reasoning, incentivizing both accuracy and calibration. Empirical evaluations across multiple QA benchmarks in factual, math, and long-form reasoning confirm its effectiveness in achieving better calibration while maintaining or improving accuracy, compared with standard RLVR and RLVR with additional classifier or probe. The paper also demonstrates that RLCR improves the self-consistency of confidence and the verbalized confidence scores of RLCR can be incorporated into test-time scaling to yield improvements in both accuracy and calibration.

**Strengths:**

* The method is simple and intuitive, and shows potential and effectiveness in improving both task accuracy and calibration.
* The reward design and the choice of score rule are well-motivated and theoretically justified.
* Uncertainty-aware reasoning and RL training for calibration is a timely and relatively under-explored topic.

**Weaknesses:**

- The paper should make it clearer the contributions and benefits compared with previous RL-based calibration methods cited in related works [1,2,3]. Given the overlap in scope, it would be very useful to include these as baselines potentially stronger than the RLVR variants. Also, given the recent debate on the contamination issue of Qwen-family models in RLVR, it would be helpful to test on other models as well.
- It would be good to isolate / disentangle the role of the Brier term and uncertainty-style CoT reasoning in improving calibration in both in-distribution and out-of-distribution settings. Is the gain mostly coming from the reward formulation (unique contribution of this paper) or simply a better confidence estimate from uncertainty reasoning?
- There is still much room to improve for the OOD settings. Given that SFT warmup is also widely-adopted, the degraded generalization performance of SFT+RLCR is a bit concerning. More insights and analyses on this would be very helpful.

---

[1]  Taming overconfidence in LLMs: Reward calibration in RLHF

[2]. LACIE: Listener-aware finetuning for calibration in large language models.

[3]. Sayself: Teaching llms to express confidence with self-reflective rationales

**Questions:**

- How sensitive are the results to the reasoning template (e.g. the analysis section, ordering)?
- For SFT warmup in Math, why not use Deepseek-R1 to output confidence scores?
- For GRPO, why remove the KL regularization?
- Are there other working scoring rules suitable for capturing uncertainty, besides Brier score?
- Is it possible to weigh the correctness and the calibration reward terms differently, and explore the tradeoff?
- Does this method show potential when there is no clear ground-truth?

---

> ### Author Response · Authors · 2025-11-22
> **Response to Reviewer xB9Y (Part 1)**
>
> Thank you for your thoughtful review. We address your comments below.
>
> ---
> **Q: Comparisons against previous RL-based calibration methods**
>
> *Ans:* We address these concerns by adding experiments with the most closely related RL-calibration baselines. **The baselines and the results are discussed in detail in point (3) of the Global Response and in Appendix I of the updated paper.**
>
> To briefly summarize the results, we find that  while all added baselines improve over standard RLVR on calibration, they consistently underperform RLCR. *RLCR continues to offer the strongest joint optimization of accuracy and calibration.*
>
>
> ---
>
> **Q: Evaluating RLCR on non Qwen2.5-family models**
>
> *Ans:* We present results comparing RLCR and RLVR on 2 new model families : **OlMo-2-7B-Instruct** and **Qwen-3-8B**. We find that RLCR has comparable accuracy to RLVR, and significantly better calibration. **We discuss these results at length in point (4) of the Global response and in Appendix F in paper.**
>
>
> ---
> **Q: Disentangling the role of the Brier Term and Uncertainty Style CoT Reasoning**
>
> *Ans:* Several reviewers raised this point. To address it clearly, **we disentangle whether RLCR’s calibration gains stem from the Brier score objective or from improved uncertainty reasoning in Point #1 of the Global Response.**
>
> As described in the global response, we find that explicit reasoning about uncertainty improves calibration for both RLCR and RLVR, but that reward-based calibration is substantially more effective than prompting alone. RLCR without Analysis nearly matches RLVR in accuracy and token cost, while dramatically improving calibration.
>
> **Please see point #1 of our global response or Section 4.6 of the paper for the full discussion!**
>
>   ---
>
> **Q: Generalization performance of SFT + RLCR**
>
> *Ans:* Several reviewers raised this point. To address it clearly, we do a deep dive to understand this accuracy drop in **Point #5 of the Global Response**, finding that the SFT+RLCR model’s O.O.D. accuracy drop actually stems not from catastrophic forgetting but from a formatting bias induced by extended RL training on math in which it reasons correctly yet inserts arbitrary numbers in the answer tags. Adding a single clarifying instruction (“be careful about what you put in the answer tags…”) to the evaluation prompt restores O.O.D. accuracy from 43.8% to 49.8%, confirming the issue is superficial misalignment rather than forgotten knowledge. Although KL regularization could mitigate such behavior, we follow recent work showing KL can hinder exploration and therefore use a KL-free GRPO variant, consistent with our focus on reward design rather than algorithmic tuning.
>
>
>
> **Please refer to Point #5 of our Global Response for more details.**
>
>   ---
>
> **Q: Sensitivity to Reasoning Template**
>
> Ans: Good question! Our design largely follows the standard practice in training reasoning models: a thinking phase followed by a final answer. We place the uncertainty analysis before the final confidence score for the same reason.
>
> In principle, the template is quite flexible. The analysis section does not need to be separate; it can be merged directly into the think step without affecting the core method. We kept them distinct to encourage explicit uncertainty reasoning, which can be useful both for users and for demonstrating our project’s intent. Likewise, while we use “answer then confidence,” asking the model to output confidence first is also a viable alternative.
>
>
>
> (In a new section 4.6 of our paper, “What Drives Calibration Gains”, we show results for RLCR without Analysis, which we show is still significantly better calibrated than both standard RLVR and RLVR with Analysis).
>
>   ---
>
> (part 2 continued below)

---

> > ### Author Response · Authors · 2025-11-22
> > **Response to Reviewer xB9Y (Part 2)**
> >
> > (continued from above)
> >
> > ---
> >
> > **Q: Using DeepSeekR1 to output confidence scores during SFT warmup for math**
> >
> > Ans: The purpose of the SFT warm-start is to expose the model to high-quality demonstrations of reasoning about uncertainty, not to inherit the final confidence values themselves. During RL training, the model learns to produce calibrated confidence scores directly from the reward signal, making explicit confidence outputs from the SFT stage unnecessary. In addition, DeepSeek-R1 performs exceptionally well on Math tasks and therefore tends to produce near-deterministic confidence values (close to 0 or 1). Such saturated scores provide limited signal for initializing a model intended to learn nuanced, graded confidence estimates.
> >
> >
> > ---
> >
> > **Q: Removing KL for GRPO**
> >
> > *Ans:* Our focus in this work is the design of the reward function; the surrounding RL algorithm is largely orthogonal. Recent work [1,2,3,4] has shown that KL regularization can suppress exploration, and it has become standard practice to run RL without KL regularization. We thus adopted a standard GRPO variant without KL regularization and intentionally refrained from extensive tuning of algorithm-specific hyperparameters.
> >
> >
> >
> > [1]: Hu, J., Zhang, Y., Han, Q., Jiang, D., Zhang, X., & Shum, H. Y. (2025). Open-reasoner-zero: An open source approach to scaling up reinforcement learning on the base model. arXiv preprint arXiv:2503.24290.
> >
> > [2]: He, J., Liu, J., Liu, C. Y., Yan, R., Wang, C., Cheng, P., ... & Zhou, Y. (2025). Skywork open reasoner 1 technical report. arXiv preprint arXiv:2505.22312.
> >
> > [3]: Zeng, A., Lv, X., Zheng, Q., Hou, Z., Chen, B., Xie, C., ... & Zhou, Z. (2025). Glm-4.5: Agentic, reasoning, and coding (arc) foundation models. arXiv preprint arXiv:2508.06471.
> >
> > [4]: Rastogi, A., Jiang, A. Q., Lo, A., Berrada, G., Lample, G., Rute, J., ... & Tang, Y. (2025). Magistral. arXiv preprint arXiv:2506.10910.
> >
> >
> > ---
> >
> > **Q: Other Scoring Rules**
> >
> > *Ans:* Yes, any proper scoring rule can be used to measure uncertainty eg., log score and spherical score.
> >
> > However, only bounded proper scoring rules can be used for RL optimization where accuracy and confidence are jointly optimized. (**Appendix E of the paper** now addresses why the Log scoring rule does not fall under the category of proper scoring rules that can be used here.) In addition to Brier, spherical score also satisfies this property. Any weighted mixture of Brier and spherical scores can also be used.
> >
> >   ---
> >
> > **Q: Weighing Correctness and Calibration reward terms**
> >
> > *Ans:* Yes! When using the Brier score, any $\lambda \geq 1$ can be used, where $\lambda$ is the weight on the binary correctness reward. We stuck to $\lambda=1$ and did not have any incentive to tweak this as the performance was matching RLVR. However, if for other models/datasets, empirically RLCR underperforms RLVR on accuracy, a simple fix is to increase $\lambda$ and give more weight to correctness!
> > Note that theoretically, tweaking this weight does not have any effect as the optimal policy for all valid $\lambda$ is the same. However, empirically it can lead to significant differences.
> >
> >   ---
> >
> > **Q: Domains without clear ground truth**
> >
> > *Ans:* In the settings we evaluate (HotpotQA and Math), a clear correctness label does exist, which allows us to quantify accuracy and compute a Brier based calibration reward. However, although outside the scope of this work, RLCR can be applicable in settings where there is not a single deterministic ground truth if we have some form of correctness signal. In settings where explicit labels are unavailable (i.e. open ended tasks), one could use implicit correctness signals, such as a rubric based LLM-as-a-judge or some sort of reward model. In these cases, the “ground truth” becomes a probabilistic correctness target rather than a single label, and the RLCR reward function reduces to optimizing calibration w.r.t. that distribution.
> >
> >
> >
> > While our paper focuses on tasks with objective correctness supervision, the framework could extend to domains without a unique ground truth through alternative correctness signals.
> >
> > ---
> >
> > Thank you again for your review. We sincerely hope these clarifications and updates address your concerns and further strengthen your confidence in the work.

---

### Official Review · Reviewer_j9oB · 2025-11-01

**Soundness:** 3
**Presentation:** 3
**Contribution:** 3
**Rating:** 6
**Confidence:** 3

**Summary:**

The paper identifies a key limitation of reinforcement learning with verifiable rewards (RLVR): the binary correctness reward penalizes equally whether the model abstains or produces an incorrect answer. Consequently, models that are initially well-calibrated tend to become overconfident after RL training. The authors propose reinforcement learning with calibration rewards (RLCR), which augments the binary correctness reward with a Brier score term based on proper scoring rules. This modification explicitly incentivizes calibrated confidence estimation in addition to correctness. RLCR substantially improves calibration both in-domain and out-of-domain without sacrificing accuracy, compared to RLVR and other post-hoc confidence baselines. Moreover, combining RLCR with confidence-weighted test-time scaling methods (e.g., weighted majority voting, best-of-N) further enhances accuracy.

**Strengths:**

- The paper tackles a well-known yet underexplored issue in RL-based reasoning: overconfidence induced by correctness-only rewards. This concern has also been raised in recent works such as *“Why Language Models Hallucinate” (OpenAI, 2025)*, but practical algorithmic remedies have been lacking.

- The proposed modification is extremely intuitive—adding a proper scoring rule (Brier score) term to the RL objective—and is supported by clean theoretical analysis showing that it jointly optimizes accuracy and calibration.

- Furthermore, the analysis of verbalized confidence self-consistency (examining both intra- and inter-solution coherence) meaningfully connects calibration to reasoning coherence.

**Weaknesses:**

1. **(Minor) Inconsistency in the definition of proper scoring rules.**
    - Equation (4) defines a proper scoring rule as one whose expected value is minimized.
    - However, the examples mix loss and reward conventions:
        - The Brier score (Eq. 6) is defined as a loss (minimized).
        - The logarithmic (Eq. 5) and spherical (Eq. 7) scores are utility functions (maximized).
    - The statement that “all these scores... are maximized” (line 144) contradicts both Eq. (4) and the Brier example.
    - This confusion between “loss” and “score” formulations creates unnecessary ambiguity and should be clarified for consistency.

2. **Lack of ablation on the necessity of “confidence analysis.”**
    - The paper claims that *chain-of-thought reasoning about uncertainty* improves calibration. However, this structured reasoning step (the `<analysis>` tag) adds considerable inference and training cost compared to simply predicting a scalar confidence value.
    - The results suggest that SFT+RLCR achieves the lowest ECE but also suffers from a substantial accuracy drop, attributed to catastrophic forgetting.
    - Therefore, it remains unclear whether the observed calibration improvement stems from enhanced uncertainty reasoning or from task-specific overfitting introduced by SFT.
    - Critically, the paper does not include RLCR results without confidence analysis (i.e., without the `<analysis>` tag), making it difficult to isolate the contribution of verbalized uncertainty reasoning to calibration performance.

**Questions:**

1. Could you report or comment on RLCR without confidence analysis (i.e., without `<analysis>` reasoning, only outputting numerical confidence)?
2. Could the authors clarify how RLCR differs from or complements traditional Best-of-N sampling (using external reward model or sentence likelihood), and whether its advantages persist when compared under identical computational budgets (number of samples)?

---

> ### Author Response · Authors · 2025-11-22
> **Response to Reviewer j9oB**
>
> Thank you for your thoughtful review. We address your comments below.
>
> ___
> **Q: Inconsistency in the definition of proper scoring rules**
>
> *Ans:* Thank you for pointing this out! We have standardized this by adopting the minimization of scoring rule definition. We have negated the log and spherical scoring rules, and changed the statement in line 144 to “minimized”. The minimization definition is also consistent with the main theorem and proof.
>
> ___
>
> **Q: Lack of ablation on the necessity of “confidence analysis.”**
>
> *Ans:* Several reviewers raised this point. To address it clearly, we disentangle whether RLCR’s calibration gains stem from the Brier score objective or from improved uncertainty reasoning in **Point #1 of the Global Response.**
>
> As described in the global response, we find that explicit reasoning about uncertainty improves calibration for both RLCR and RLVR, but that reward-based calibration is substantially more effective than prompting alone. RLCR without Analysis nearly matches RLVR in accuracy and token cost, while dramatically improving calibration.
>
>
> ---
>
> **Q: Accuracy drop of SFT + RLCR**
>
> *Ans:* Several reviewers raised this point. To address it clearly, we do a deep dive to understand this accuracy drop in **Point #5 of the Global Response**, finding that the SFT+RLCR model’s O.O.D. accuracy drop actually stems not from catastrophic forgetting but from a formatting bias induced by extended RL training on math in which it reasons correctly yet inserts arbitrary numbers in the answer tags. Adding a single clarifying instruction (“be careful about what you put in the answer tags…”) restores O.O.D. accuracy from 43.8% to 49.8%, confirming the issue is superficial misalignment rather than forgotten knowledge. Although KL regularization could mitigate such behavior, we follow recent work showing KL can hinder exploration and therefore use a KL-free GRPO variant, consistent with our focus on reward design rather than algorithmic tuning.
>
>
> **Please refer to Point #5 of our Global Response for more details.**
>
>   ---
>
>
> **Q: How RLCR differs from traditional Best of N sampling**
>
> *Ans:* Test-time scaling is one of the possible applications of RLCR. The main insight is that in settings where external reward models are not available or unreliable, the model’s verbalized confidence can be used as reward in Best-of-N sampling.
>
>
>
> In settings where a good reward model is available and easy to integrate, traditional Best-of-N sampling with external RM is likely more powerful, although verbalized confidence can still complement it through simple aggregation. For example, the reward of a generation could be a weighted score of its own verbalized confidence and the external RM score.
>
>
>
> To compare against sentence likelihood (available for open models but not frontier models), we add 2 new baselines to our accuracy test-time scaling plot (**Figure 3A of the paper**):
>
> 1.  **Max Gen Likelihood:** As suggested by the reviewer, we do Best-of-N sampling using the average likelihood of the generation as reward. $R = \dfrac{p(y|x)}{|y|}$.
>
> 2.  **Gen Likelihood-Weighted Maj. Vote:** We combine majority voting with the likelihood-based reward scores, similar to our confidence-weighted majority vote.
>
>
>
>
> *From these results, we see that while Best-of-N sampling using sentence likelihood improves performance with more computation, it lags behind the verbalized confidence-based scaling methods powered by RLCR.*
>
>
>
> We are also happy to add external RM-based scaling baselines to the plot, but believe that it is not central to our paper.
>
>   ____
>
> Thank you again for your review. We sincerely hope these clarifications and updates address your concerns and further strengthen your confidence in the work.

---

### Official Review · Reviewer_ffd5 · 2025-11-01

**Soundness:** 2
**Presentation:** 3
**Contribution:** 2
**Rating:** 4
**Confidence:** 4

**Summary:**

- This paper proposes a reward function that simultaneously incentivizes both answer correctness and probabilistic calibration, moving beyond binary correctness signals traditionally used in RLHF. By introducing a smooth, bounded Brier-score term, the method encourages models to be not only accurate but also appropriately uncertain. The proposed approach, RLCR (Reinforcement Learning with Calibrated Rewards), addresses common issues like reward hacking found with log-likelihood and unbounded scoring rules. The authors demonstrate that RLCR improves both calibration and output quality across reasoning tasks. Extensive experiments show that RLCR outperforms binary reward baselines in terms of calibration error, and answer diversity. Notably, the method avoids overconfident hallucinations and penalizes low-confidence correct answers, promoting reliable confidence estimates.

**Strengths:**

- The paper tackles a timely and practically-relevant problem supported by a fair amount of experiments.
- Overall, the paper is clearly written and easy to follow.

**Weaknesses:**

- **Model coverage** The paper only tests Qwen2.5-7B, and additional models, maybe a non-reasoning model, should be included to demonstrate that RLCR generalizes beyond a single model. I must state that this is not a mere superficial comment rather one of my major concerns.
- **Fair evaluation** Page 27 notes that RLCR produces shorter `<think>` sections than RLVR. It is unclear whether the results were evaluated under equal inference budgets, and this should be explicitly specified to ensure fairness. Further, I wonder whether reasoning length correlates with uncertainty.
- **Missing hyperparameters** Key training details are absent, including GPU type, learning rate, and GRPO training configuration.
- **Oracle-based labeling concern** For Math datasets, the uncertainty labels are generated using DeepSeek-R1, a far larger model. This could transfer oracle-level knowledge and confound the evaluation.  Please correct me if I understood something wrong.
- **Inconsistent results and missing discussion** While Table 1 reports averaged results, some findings contradict this trend. For example, in Appendix E.3 (Table 3), RLVR achieves an ECE of 0.09 whereas RLCR records 0.30 with nearly identical accuracy. The paper should analyze and discuss such failure cases. Likewise, lines 356–357 state that RLCR maintains accuracy, but it clearly lags behind RLVR.
- **Formatting confound (analysis tag)** RLCR uses an `<analysis>` tag that allows longer uncertainty reasoning, whereas baselines do not. This formatting difference must be ablated by testing RLCR without the analysis section so that the only change is the reward function. This is crucial because (1) RLCR incurs higher inference costs (see p. 24, 26), and (2) prior work (Yoon et al., 2025) already shows that extended uncertainty reasoning improves calibration. Further, a fair comparison should thus include *RLVR + uncertainty analysis* as baselines.
- **Missing baselines** The paper omits comparisons with prior works that enhance verbalized uncertainty. Even if those do not jointly optimize accuracy and calibration via RL, they should be included for context. Moreover, instead of training separate RLVR classifiers, a more natural baseline would be SFT with uncertainty labels (e.g., oracle- or self-consistency-based) alongside existing RL-calibration methods.
- **Failure cases for unbounded scoring rules** Table 1 shows BCE classifiers outperforming Brier ones, yet there is no empirical support for the theoretical claim that unbounded proper scores lead to overfitted low-confidence answers. Experiments using other scoring rules such as logarithmic (Eq. 5) or spherical (Eq. 7) are needed.
- **Catastrophic forgetting and missing regularization** In Table 1(b), SFT + RLCR performs worse, and the authors blame this on catastrophic forgetting. It is unclear why KL-regularization—a rather straightforward fix—was not applied.
- **Figure 3 interpretation** Figure 3 indicates (1) some incorrect answers appear with high confidence and (2) majority voting consistently underperforms confidence-weighted voting by a small margin. Clarify: (a) how majority voting handles ties; and (b) include plots of both accuracy and frequency (y-axis) versus confidence bins (x-axis) to visualize calibration.

**Questions:**

- **Definition of p** Lines 92 and 134 use $p$ without definition; it should be mentioned that $p$ is the true distribution. In line 145, I doubt that the true probability $p(a=1)$ should be $p(a)$.
- **GRPO acronym** Line 202 should spell out GRPO and include a citation.
- **Confidence scale inconsistency** Appendix B.4 defines confidence in [0, 100] at first, then [0, 1]. I wonder why this is the case.
- **Table 1 labeling** Upper panel uses “OOD” while the lower uses “OOD averaged”; these two should match.
- **Dataset choice** Why use *hotpotqa-distracted*? Was the original HotpotQA too easy (near-perfect accuracy/calibration)?
- **ECE binning** What happens if the number of bins increases (e.g., 20)?
- **AUROC definition** Clarify whether AUROC is computed from verbalized confidence or token-level probabilities.

---

> ### Author Response · Authors · 2025-11-22
> **Response to Reviewer ffd5 (Part 1)**
>
> Thank you for your thoughtful review. Below, we address your individual concerns.
>
>
> ___
>
>
>
>
> **Q. More models should be included to demonstrate that RLCR generalizes beyond a single model.**
>
> *Ans.* We present results comparing RLCR and RLVR on 2 new model families : **OlMo-2-7B-Instruct** and **Qwen-3-8B.** We find that RLCR has comparable accuracy to RLVR, and significantly better calibration. **We discuss these results at length in point (4) of the Global response and in Appendix F in paper.**
>
>   ___
>
> **Q. It is unclear whether the results were evaluated under equal inference budgets, and this should be explicitly specified to ensure fairness.**
>
> *Ans.* All methods were evaluated under the same maximum token budget (4096). While RLVR naturally produces shorter CoTs than RLCR, Section 4.6 shows that RLCR w/o Analysis generates roughly the same number of tokens as RLVR, yet matches RLVR in accuracy and substantially outperforms it in calibration. Thus, when token efficiency is important, RLCR w/o Analysis offers a practical alternative that preserves most (marginally reduced calibration) of the benefits of RLCR without the additional token cost.
>
>
> **We also discuss these results in point (1) of the Global response and in Section 4.6 of the paper.**
>
>   ___
>
> **Q. Missing training hyperparameters need to be added.**
>
> *Ans.* We have added relevant training parameters in Appendix B.2.
>
> ___
>
> **Q. For Math datasets, the uncertainty labels are generated using DeepSeek-R1, a far larger model. This could transfer oracle-level knowledge and confound the evaluation.**
>
> *Ans.* Thank you, this is a valid question. We only obtain uncertainty reasoning from Deepseek-R1, which is used to warmstart the `<analysis>`. We do not generate solutions from R1 nor do we ask it to output confidence labels, as the confidence labels will be biased (because Deepseek-R1 is extremely good at Math, it will always have 0 or 1 as confidence for the solutions it is provided with).
>
> While the uncertainty reasoning can itself transfer oracle-level knowledge, we find minimal evidence of this:
>
> 1.  The accuracies of all 3 models (RLVR, RLCR and SFT+RLCR) are roughly the same.
>
> 2.  We explicitly instruct R1 to not attempt to solve the question, instead only asking it to reason about uncertainty in a given solution (the solution itself is generated through the 7B model).
>
> 3.  The SFT warmup is relatively light (500 datapoints).
>
>
>
> ___
> **Q. Table 1 reports averaged O.O.D. results, but some findings contradict this trend. For example, in Appendix E.3 (Table 3), RLVR achieves an ECE of 0.09 whereas RLCR records 0.30 with nearly identical accuracy. The paper should analyze and discuss such failure cases.**
>
> *Ans.* Thank you,  we have added a discussion on the failure case mentioned by the reviewer in **Appendix L** .
>
> The failure case mentioned is on CommonsenseQA, where all methods have roughly ~90% accuracy. The RLVR baseline is extremely overconfident, and blindly reports 85-100% confidence on every question and every dataset. On CommonsenseQA, this overconfidence actually leads to great calibration because it aligns with the accuracy on the dataset. The great performance of RLVR is thus a spurious correlation. However, we also agree that RLCR performs poorly on this specific dataset and has extremely underconfident predictions, which we briefly discuss in the Appendix as well.
>
> ___
> **Q. RLCR uses an `<analysis>` tag that allows longer uncertainty reasoning, whereas baselines do not. This formatting difference must be ablated by testing RLCR without the analysis section so that the only change is the reward function.**
>
> *Ans.* Thank you for this insightful question, addressing it substantially improved the paper. We added a new experiment to disentangle the contributions of the calibration-aware reward and the explicit uncertainty reasoning. **This experiment is discussed in detail in point (1) of the Global Response and in Section 4.6 of the updated paper.**
>
> To briefly summarize the results, we evaluate *RLCR without the analysis section* and *RLVR with the analysis section*, comparing them against the original RLCR and RLVR. We find that both components—the reward function and the explicit uncertainty reasoning—independently contribute to improved performance.
>
>
> ---
>
> Our response is continued in the comment below.

---

> ### Author Response · Authors · 2025-11-22
> **Response to reviewer ffd5 (Part 2)**
>
> (continued from above)
>
> ---
> **Q. The paper omits comparisons with prior works that enhance verbalized uncertainty. Even if those do not jointly optimize accuracy and calibration via RL, they should be included for context.**
>
> *Ans.* We address these concerns by adding experiments with the most closely related RL-calibration baselines. **The baselines and the results are discussed in detail in point (3) of the Global Response and in Appendix I of the updated paper.**
>
>
> To briefly summarize the results, we find that  while all added baselines improve over standard RLVR on calibration, they consistently underperform RLCR. RLCR continues to offer the strongest joint optimization of accuracy and calibration.
>
> ---
>
> **Q. There is no empirical support for the theoretical claim that unbounded proper scores (such as log loss) can lead to reward hacking.**
>
> *Ans.* Thank you for the suggestion. We address this by adding a new appendix that: (a) provides an intuitive explanation of the differences between log and Brier scoring rules, (b) presents a toy experiment demonstrating that combining correctness with log score can lead to reward hacking while Brier does not, and (c) reports results on HotpotQA showing that log score marginally underperforms Brier score, although we do not observe hacking in this setting.
>
>
>
> **These experiments are discussed in detail in point (2) of the Global Response and in Appendix E of the updated paper.**
>
>
> ---
>
> **Q. In Table 1(b), SFT + RLCR performs worse, and the authors hypothesize catastrophic forgetting to be the culprit. Further analysis would be helpful. Why was KL regularization not used?**
>
> *Ans.* Thank you, this is a valid concern. **We have  discussed this in detail in point (5) of the Global Response and in Appendix J of the updated paper.**
>
> After deeper analysis on the SFT+RLCR model, we found that the root cause of the poor performance was a form of Math overfitting that caused the model to frequently output numbers in its `<answer>` (even for factual questions). We found that adding a single line to the system prompt (asking it to not arbitrarily put numbers inside the answer tags) increases accuracy from 43.8% to 50%, which is comparable to other baselines. These results show that the model did not actually forget information, but became prone to output numbers as its answer. **We did not observe this abnormality in the other models/baselines.**
>
>
>
>   ---
>
> **Q. Figure 3 indicates (1) some incorrect answers appear with high confidence and (2) majority voting consistently underperforms confidence-weighted voting by a small margin. Clarify: (a) how majority voting handles ties; and (b) include plots of both accuracy and frequency (y-axis) versus confidence bins (x-axis) to visualize calibration.**
>
> *Ans.*
>
> (1) Majority voting handles ties by random choice between the answers with the highest number of votes.
>
> (2) We generated calibration charts overlaid with confidence-frequency histograms in Figures 8 and 9 of Appendix H.1. We create these plots for RLVR, RLCR and RLVR+Classifier, both in-distribution and O.O.D. We find that RLCR is well-calibrated and distributes confidences over a wide range of values. In contrast, the confidences of RLVR cluster closely around 1.
>
>
> ---
> **Q: Definition of p**
>
> Ans: Thank you for pointing this out. Near lines 92 and 134, we have clarified that p is the true underlying probability distribution over correctness labels.
>
>
> ---
> **Q: GRPO Acronym - Line 202 should spell out GRPO and include a citation.**
>
> Ans: We have made this adjustment, thank you for pointing this out!
>
> ---
> **Q: Confidence scale inconsistency - Appendix B.4 defines confidence in [0, 100] at first, then [0, 1]. I wonder why this is the case.**
>
> Ans: We found that baselines which have not been trained to output verbalized confidence scores (such as RLVR) can struggle with the [0,1] scale. We instead prompt them with the 0-100 scale, which is more intuitive and leads to marginally better performance.
>
> ---
>
> (response continued below)

---

> > ### Author Response · Authors · 2025-11-22
> > **Response to Reviewer ffd5 (Part 3)**
> >
> > (continued from above)
> >
> > ---
> > **Q: Table 1 labeling - Upper panel uses “OOD” while the lower uses “OOD averaged”; these two should match.**
> >
> > Ans: We have made this adjustment, thank you for pointing this out!
> >
> > ---
> > **Q: Dataset choice - Why use hotpotqa-distracted? Was the original HotpotQA too easy (near-perfect accuracy/calibration)?**
> >
> > Ans: Thank you for the question. We use HotpotQA-Distracted because it provides a substantially more challenging and uncertainty-sensitive environment than the Standard hotpotQA setting. HotpotQA-Distracted introduces 8 additional distracted paragraphs containing irrelevant information, increasing context length, noise, and multi-hop interference. This makes the reasoning task more realistic and substantially harder, especially for confidence estimation.
> >
> >
> >
> > Beyond this, we further modify the dataset by intentionally removing 0, 1, or 2 paragraphs that contain key supporting information. Standard HotpotQA provides all required evidence for multi-hop reasoning, so models typically have access to everything they need. By selectively removing supporting paragraphs, we create a controlled setting where the available evidence is incomplete, which introduces a new kind of informational uncertainty that models often encounter in standard QA settings. Finally, we report results on both the original HotpotQA-Distracted as well as our modified HotpotQA.
> >
> > ---
> > **Q: ECE binning: What happens if the number of bins increases (e.g., 20)?**
> >
> > Ans: We ran this and found a negligible change (~0.01) in ECE when the number of bins is changed to 20. We are happy to report the full comparative numbers if needed.
> >
> >
> > ---
> > **Q: AUROC definition: Clarify whether AUROC is computed from verbalized confidence or token-level probabilities.**
> >
> > Ans: AUROC is computed from verbalized confidence.
> >
> > ---
> >
> >
> > Thank you again for your review. We sincerely hope these clarifications and updates address your concerns and strengthen your confidence in the work.

---

> > > ### Comment · Reviewer_ffd5 · 2025-11-27
> > >
> > > Thank you for the detailed rebuttal, and I have gone through all the newly added experimental results. The new experiments on additional models, the ablations disentangling the effects of the analysis prompt, the added comparisons to RL-based calibration baselines, etc., convincingly addressed most of my main concerns. Overall, I believe the paper has been significantly strengthened, and I am thus updating my score from 4 to 6, leaning toward acceptance.

---

### Author Response · Authors · 2025-11-22
**Global Response (Part 1)**

### ***Global Response to Reviewers***

We thank the reviewers for their thoughtful and constructive feedback. In response, we conducted a significant number of experiments that we believe have greatly strengthened the paper.

These new results:
- clarify the distinct contributions of Brier-score–based rewards and uncertainty reasoning
- provide empirical evidence contrasting Brier and log scoring
- add comparisons to the most relevant RL-based calibration baselines
- demonstrate that RLCR generalizes across model families
- and explain the accuracy drop observed in SFT+RLCR.

_____
### **1. Disentangling Brier Score and Uncertainty Reasoning**

Several reviewers (ffd5, j9ob, xB9Y) correctly pointed out that RLCR is prompted to reason about its uncertainty, whereas RLVR is not, raising the question: *Where do RLCR’s calibration gains actually come from?* Are they because of our proposed reward structure, or simply because of the explicit uncertainty reasoning performed by RLCR?


To isolate the contribution of these components, we ablated the uncertainty-reasoning portion of the CoT prompt and evaluate the following four variants (detailed results in **Section 4.6** of the paper):

1.  **RLCR:** Standard RLCR with both the calibration-aware reward and explicit uncertainty reasoning.

2.  **RLCR w/o Analysis:** RLCR evaluated without the uncertainty-reasoning section. At inference time, the model is instructed to output only `<think>,` `<answer>,`and `<confidence>,`.The model is instructed to not perform any uncertainty reasoning.

3.  **RLVR:** Standard RLVR baseline

4.  **RLVR w/ Analysis:** RLVR evaluated with the same uncertainty-reasoning prompt used for RLCR, enabling it to perform identical uncertainty analysis.




Results comparing accuracy, calibration and token count of the above variants are presented in **Table 2 of the paper (Section 4.6)**. These results reveal several new key findings:



1.  **Explicit uncertainty reasoning improves calibration for both RLCR and RLVR:** Across both training and OOD settings, the variants with uncertainty reasoning achieve better calibration than their no-analysis counterparts.

2.  **Reward-based calibration is substantially more effective than prompting alone:** Although RLVR w/ Analysis improves over vanilla RLVR, it remains far behind both RLCR variants, including RLCR w/o Analysis. Thus, prompting a model to reason about uncertainty provides only modest gains compared to explicitly training with a calibrated reward.

3.  **RLCR w/o Analysis nearly matches RLVR in accuracy and token cost, while dramatically improving calibration:** Standard RLCR outputs significantly more tokens than RLVR due to uncertainty reasoning. However, in settings where token efficiency is a concern, RLCR w/o Analysis offers a drop-in alternative that preserves accuracy and token efficiency of RLVR while significantly improving calibration.




**Overall, these results show that both components—explicit analysis and calibration-aware reward learning—contribute to improved calibration.**




_____
### **2. Empirical Differences between Log and Brier Scores**

Two reviewers (ffd5 and bCz3) suggested that although we provide theoretical results on why log scoring rule does not have the correctness incentive, adding empirical support for these insights will strengthen the paper. In **Appendix E**, we do a deeper theoretical and empirical dive into differences between Brier and log score when jointly optimizing accuracy and calibration:



1.  We add a graphical/intuitive argument showing the differences in expected reward between log score and Brier scores.

2.  We then present results of a Qwen-7B model on a toy arm experiment. Here, we find that using log scoring rule leads to reward hacking: The model learns to output incorrect/invalid answers and with 0 confidence, leading to 0 accuracy but perfect calibration. In contrast, this does not happen when using Brier scores! The model outputs valid answers, and tries to maximize accuracy and calibration together. These results validate our theoretical predictions.

3.  Finally, we run the Qwen-7B model with the log scoring rule on our HotpotQA dataset. Here, we find that although log scoring marginally underperforms on both accuracy and calibration compared to RLCR, there do not seem to be any signs of reward hacking.




**Overall, these results show that even though using log scoring can lead to reward hacking (as shown in the toy experiment), for many datasets hacking might not happen in practice. We believe the emergence of hacking is dependent on both the data distribution as well as the model size.**


_____
Global points 3, 4, and 5 are continued below.

---

> ### Author Response · Authors · 2025-11-22
> **Global Response (continued, part 2)**
>
> (continued from above)
>
> _____
> ### **3. Missing Baselines**
>
>
> Two reviewers (ffd5 and xB9Y) requested comparisons to recent RL-based calibration methods to better contextualize how RLCR relates to prior work. In **Appendix F**, we address these concerns by adding experiments with the most closely related RL-calibration baselines:
>
>
>
> 1.  **Calibration-Only RL** [1,2]: We add two calibration-only RL variants from prior work—one that applies the Brier reward to the full sequence and another that applies it only to the analysis and confidence sections. Because these methods do not natively optimize accuracy, we initialize them from the trained RLVR model (which has high accuracy) to ensure a fair comparison.
>
> 2.  **Abstention-RL** [3,4]: We include abstention-based RL methods that reward explicit “I don’t know’’ responses. The reward structure $+1$ for correct answers, $0$ for incorrect answers, and an intermediate reward $\lambda$ for explicit abstentions (e.g., “I don’t know”). We use $\lambda=0.5$. Because abstention models do not produce confidence scores, we evaluate calibration by using a test-time prompt that instructs the model to *never* abstain.
>
>
>
>
> Full results in **Appendix F** show that while all added baselines improve over standard RLVR on calibration, they consistently underperform RLCR. Full-sequence calibration RL collapses to a degenerate policy that outputs incorrect answers with zero confidence, the analysis-only variant maintains accuracy but achieves weaker calibration, and abstention-RL underperforms on calibration because it never learns fine-grained confidence estimates.
>
>
> **Overall, these results show that RLCR provides the strongest joint optimization of accuracy and calibration.**
>
>
>
> [1]: Stangel, P., Bani-Harouni, D., Pellegrini, C., Özsoy, E., Zaripova, K., Keicher, M., & Navab, N. (2025). Rewarding Doubt: A Reinforcement Learning Approach to Calibrated Confidence Expression of Large Language Models. arXiv preprint arXiv:2503.02623.
>
>
>
> [2]: Xu, T., Wu, S., Diao, S., Liu, X., Wang, X., Chen, Y., & Gao, J. (2024). Sayself: Teaching llms to express confidence with self-reflective rationales. arXiv preprint arXiv:2405.20974.
>
>
>
> [3]: Wei, Z., Yang, X., Sun, K., Wang, J., Shao, R., Chen, S., ... & Dong, X. L. (2025). TruthRL: Incentivizing truthful LLMs via reinforcement learning. arXiv preprint arXiv:2509.25760.
>
>
>
> [4]: Mohamadi, M. A., Wang, T., & Li, Z. (2025). Honesty over Accuracy: Trustworthy Language Models through Reinforced Hesitation. arXiv preprint arXiv:2511.11500.
>
>
>
>
>
> _____
> ### **4. Generalization to other model families**
>
>
> Two reviewers (ffd5 and x9BY) requested that we demonstrate empirical results for RLCR on model families beyond Qwen2.5. In **Appendix F**, we present results comparing RLCR and RLVR on 2 new model families:
>
>
>
> 1.  **OlMo-2-7B-Instruct** [1]: We provide results of RL training on a commonly used instruct model. The trends mirror our main results: RLCR and RLVR achieve comparable accuracy, both outperform the base model, and RLCR delivers significantly stronger calibration.
>
> 2.  **Qwen-3-8B** [2]:The Qwen-3 series of models implement long CoT thinking by default. Again, we find the same trend holds. RLVR and RLCR are comparable in accuracy, while RLCR is significantly better-calibrated.
>
>
>
>
> **Overall, these experiments demonstrate that RLCR’s reward design and uncertainty-reasoning mechanism generalize well across model families, consistently improving calibration without sacrificing accuracy.**
>
>
>
>
> [1]: OLMo, T., Walsh, P., Soldaini, L., Groeneveld, D., Lo, K., Arora, S., ... & Hajishirzi, H. (2024). 2 OLMo 2 Furious. arXiv preprint arXiv:2501.00656.
>
>
>
> [2]: Yang, A., Li, A., Yang, B., Zhang, B., Hui, B., Zheng, B., ... & Qiu, Z. (2025). Qwen3 technical report. arXiv preprint arXiv:2505.09388.
>
> _____
> Point 5 continued below.

---

> > ### Author Response · Authors · 2025-11-22
> > **Global Response (part 3)**
> >
> > (continued from above)
> >
> > _____
> > ### **5. Analyzing the accuracy drop of SFT+RLCR**
> >
> > In original results, we found that our SFT+RLCR model (trained on Math) experienced a substantial drop in O.O.D accuracy. While RLVR and vanilla RLCR achieve O.O.D. accuracies of 52.5\% and 50.9\% respectively, SFT+RLCR reaches only 43.8\%. Multiple reviewers (ffd5, x9BY, j9ob) expressed that this deserved further investigation.
> >
> > To understand this accuracy drop, we start examining the O.O.D outputs of the SFT+RLCR model and found an abnormality unique to the SFT+RLCR model: the model often identifies the correct answer during its reasoning, but then places an incorrect, seemingly random number inside the answer tags, leading to the answer being marked as incorrect. The extended RL training on Math seemed to have induced some form of overfitting on this model.
> >
> > To test whether the reason behind the accuracy drop was catastrophic forgetting or a more superficial misalignment, we reran the evaluation with a slightly modified prompt with an additional instruction:
> >
> > `Be careful about what you put in the answer tags. Do not arbitrarily put numbers there if the question has nothing to do with Math.`
> >
> > **Remarkably, this simple change improves O.O.D.accuracy from 43.8\% to 49.8\% (**Appendix J** has a full comparative analysis of the results)! This indicates that the accuracy degradation was not due to catastrophic forgetting, but rather due to formatting biases learning during heavy Math training.**
> >
> > In general, adding KL regularization might be able to prevent this behavior, which was also pointed out by some reviewers. However, our central focus in this work was the design of the reward function; the surrounding RL algorithm is largely orthogonal. Recent work [1, 2, 3, 4] has shown that KL regularization can suppress exploration, and it has become standard practice to run RL without KL regularization. We thus adopted a GRPO variant without KL regularization and intentionally refrained from extensive tuning of algorithm-specific hyperparameters.
> >
> > [1]: Hu, J., Zhang, Y., Han, Q., Jiang, D., Zhang, X., & Shum, H. Y. (2025). Open-reasoner-zero: An open source approach to scaling up reinforcement learning on the base model. arXiv preprint arXiv:2503.24290.
> >
> > [2]: He, J., Liu, J., Liu, C. Y., Yan, R., Wang, C., Cheng, P., ... & Zhou, Y. (2025). Skywork open reasoner 1 technical report. arXiv preprint arXiv:2505.22312.
> >
> > [3]: Zeng, A., Lv, X., Zheng, Q., Hou, Z., Chen, B., Xie, C., ... & Zhou, Z. (2025). Glm-4.5: Agentic, reasoning, and coding (arc) foundation models. arXiv preprint arXiv:2508.06471.
> >
> > [4]: Rastogi, A., Jiang, A. Q., Lo, A., Berrada, G., Lample, G., Rute, J., ... & Tang, Y. (2025). Magistral. arXiv preprint arXiv:2506.10910.
> >
> >
> > _____

---

### Author Response · Authors · 2025-12-03
**Final Author Response**

Dear AC/Reviewers

We sincerely appreciate your service, and genuinely believe that the reviewing process has significantly increased the strength of our paper. Below, we summarize the paper, initial reviews and the current state of the rebuttal.

**Our paper introduces RLCR, RL–based framework that jointly optimizes correctness and confidence using a proper scoring-rule reward and explicit uncertainty reasoning. This approach is novel because it provides the first practical RL method that directly targets calibration without sacrificing accuracy, supported by strong theoretical guarantees.**

The initial reviews were generally positive (**initial scores 4/4/6/6**) and all reviewers grasped the key strengths of the work: **Timely  problem** (Reviewer ffd5,j90B, xB9Y), **well-written** (Reviewer ffd5,bCz3) , **intuitive** (Reviewer j90B, xB9Y) and **theoretically justified** (Reviewer j90B, xB9Y,bCz3).

The reviewers believed that the paper could be further strengthened with more experiments:

1.  **Disentangling Brier Score and Uncertainty Reasoning (Reviewer ffd5, j9ob, xB9Y): In RLCR, models are 1) trained with our proposed proper scoring-rule reward function and 2) prompted to reason about their uncertainty. This raised the question: Where do RLCR’s calibration gains actually come from?**
A.  We added a new experiment to disentangle the two components. We find that both components—the reward and the explicit uncertainty reasoning—independently contribute to improved performance. **( see point (1) of the Global Response,  Section 4.6 of the updated paper).**

----

2.  **Empirical differences between Log and Brier scores (Reviewer ffd5, bCz3): Two reviewers suggested that although we provide theoretical results on why log scoring rule does not have the correctness incentive, adding empirical support will strengthen the paper.**
A.  We address this by adding a new appendix that compares log vs.Brier scoring rules, includes a toy example illustrating that log score can induce reward hacking, and presents HotpotQA results showing that log score does not exhibit hacking in one of the practical settings. **(see point (2) of GR,  Appendix E of paper).**

----

3.  **More Baselines (Reviewer ffd5, xB9Y): Two reviewers requested comparisons to recent RL-based calibration methods.**
A.  We added experiments with closely related RL-calibration baselines. We find that while all added baselines improve over standard RLVR on calibration, they consistently underperform our method. **(see point (3) of GR,  Appendix I of paper).**

----

4.  **Generalization to other model families(Reviewer ffd5, xB9Y): Two reviewers requested that we demonstrate empirical results for RLCR on model families beyond Qwen2.5.**
A.  We present results comparing RLCR and RLVR on 2 new model families. We find that our method generalizes well to these model families matching RLVR in accuracy and outperforming it on calibration. **(see point (4) of GR,  Appendix F of paper).**

----

5.  **Analyzing accuracy drop of SFT+RLCR (Reviewer ffd5, xB9Y, j9oB): In original results, we found that our SFT+RLCR model (trained on Math) experienced a substantial drop in O.O.D accuracy. Reviewers wanted us to dig deeper into this.**.
A.  We found an abnormality unique to our SFT+RLCR model: a form of Math overfitting that caused the model to frequently output numbers in its (even for factual questions). We found that adding a single line to the system prompt (asking it to not arbitrarily put numbers inside the answer tags) recovers most of O.O.D accuracy. **(see point (5) of GR, Appendix J of paper).**

----

6.  **Analyzing confidence distributions (Reviewer ffd5, bCz3): Reviewers wanted to understand if the model outputs a diverse range of confidences.**
A. We generated calibration charts overlaid with confidence-frequency histograms. We find that R_LCR exhibits a healthy spread of confidence values and is well-calibrated. **(see Appendix H).**

----


### **Overall, we conducted more than 6 sets of experiments, which we believed have greatly increased the strength of the paper. We have compared against 3 new baselines, showed generalization to 2 new model families, conducted 2 new ablation studies and added 2 new analysis subsections.**

### **Our original scores were 4/4/6/6**. Only reviewer ffd5 engaged was able to respond in time during the discussion phase and **raised their score from 4 to 6**, and acknowledged that the rebuttal *“convincingly addressed most of their concerns”*. **This raised the average rating of our paper from 5 to 5.5**. We would also like to highlight that reviewer ffd5 had the broadest set of concerns. In fact, the primary concerns of reviewers j9oB and xB9Y were a subset of reviewer ffd5’s concerns. We are confident that addressing reviewer ffd5’s concerns is a strong indicator that other reviewers would also have reacted positively if the discussion phase would have proceeded normally.

---

### Meta-Review · Area_Chair_UkKp · 2025-12-31

**Summary:**

This paper proposes to optimize performance and calibration of LLMs through reinforcement learning. The approach is simple and well-motivated. The reviewers raised concerns about (1) experimental comparisons to RL-based calibration methods, (2) disentangling the role of the brier term and uncertainty-style CoT reasoning, (3) reduced performance in OOD evaluation, and (4) experiments with other LLM model families, and (5) further analysis on the confidence distribution. The authors thoroughly addressed these concerns.

In addition to the reviewer comments, I have a couple further suggestions for improving the paper:
1. It would be interesting to include common selective classification metrics, e.g. the average accuracy at a particular confidence level, to see if the improvements in calibration translate into improved selective classification (i.e. for applications where mistakes are far more costly than abstaining)
2. There are a few (published) related works that use RL for improving factuality / reducing hallucination that seem worth discussing in the related work section:
    * Reasoning Models Hallucinate More: Factuality-Aware Reinforcement Learning for Large Reasoning Models (https://arxiv.org/abs/2505.24630)
    * Fine-tuning Language Models for Factuality (https://arxiv.org/abs/2311.08401)
    * FLAME : Factuality-Aware Alignment for Large Language Models (https://arxiv.org/abs/2405.01525)

Overall, I recommend accept.

**Reviewer Concerns:**

See summary.

**Reviewer Scores:**

I believe the reviewers would have increased their scores.

---

### Decision · Program_Chairs · 2026-01-26

Accept (Poster)